# Real-time label-free imaging of living crystallization-driven self-assembly

Yujie Guo [1,4], Tianlai Xia [2,4], Vivien Walter [3], Yujie Xie [2], Julia Y. Rho[2], Laihui Xiao [2], Rachel K. O'Reilly [2] ✉ & Mark I. Wallace [1] ✉

Living crystallization-driven self-assembly (CDSA) of semicrystalline block copolymers is a powerful method for the bottom-up construction of uniform polymer microstructures with complex hierarchies. Improving our ability to engineer such complex particles demands a better understanding of how to precisely control the self-assembly process. Here, we apply interferometric scattering (iSCAT) microscopy to observe the real-time growth of individual poly($\varepsilon$-caprolactone)-based fibers and platelets. This label-free method enables us to map the role of key reaction parameters on platelet growth rate, size, and morphology. Furthermore, iSCAT provides a contrast mechanism for studying multi-annulus platelets formed via the sequential addition of different unimers, offering insights into the spatial distribution of polymer compositions within a single platelet.

Monodisperse nanomaterials with precise morphologies are a vital component of many recent nanotechnologies[1,2]. One promising synthetic route to prepare particles with defined non-spherical architectures is the crystallization-driven self-assembly (CDSA) of block copolymers[3–9]. Living CDSA provides access to complex hierarchical nanostructures with impressive morphological and dimensional control; and this complexity enables wide-ranging applications in drug delivery[10–13], colloid stabilization[14,15], catalysis[16–18], optoelectronics[19,20], and information storage[21–23]. The livingness of CDSA is achieved by initiating growth from uniform seed crystallites, which provide the nucleation that enables self-assembly to start from a common point (Fig. 1A)[6,8,10,24–26]. Realizing the full potential of CDSA for nanostructure synthesis requires a precise understanding of how CDSA particles grow and how this self-assembly process can be controlled.

Most previous reports monitoring the self-assembly of block copolymers derive kinetics from ensemble properties: For example, light scattering can be employed to report on the average size of CDSA micelles[27–29]. However, these ensemble methods are insensitive to heterogeneity in the growth of individual nanoparticles and provide no direct information on particle morphology. Given the uniform complex hierarchical structures desired by CDSA, this ensemble averaging is a significant drawback in understanding and quantifying growth

mechanics. Thus, a key hurdle to properly understanding CDSA lies in providing methods capable of revealing how assembly is controlled on a particle-by-particle basis.

Transmission electron microscopy (TEM) is arguably the most common method used to study the size evolution of individual polymer assemblies[7,28,30–32]. However, conventional TEM provides only snapshots of the distribution of particle sizes during growth, lacking the capability to track the transformations of individual nanoparticles. Exciting recent advances in liquid cell TEM have opened a route to monitor the liquid phase reaction in situ with millisecond to second temporal resolution, albeit at the expense of significant instrumental complexity[33–35]. Solution-phase atomic force microscopy (AFM) has also been applied to study the mechanism of interfacial seeded-growth of 1D micellar nanoparticles. However, tip-induced nanofiber fragmentation has been reported as a drawback by creating new active interfaces and affecting kinetics[36]. Fluorescence microscopy offers a perhaps less invasive alternative, but at the expense of spatial resolution: For example, confocal laser scanning microscopy (CLSM) has been used to visualize 1D fiber growth at 100 ms temporal resolution with 120 nm lateral spatial resolution[28]. Super-resolution fluorescence microscopy techniques can improve these limits with, for instance, 76 nm precision and 15 ms temporal resolution of the growth of 1D

[1]Department of Chemistry, King's College London, London, UK. [2]School of Chemistry, University of Birmingham, Birmingham, UK. [3]Department of Engineering, King's College London, London, UK. [4]These authors contributed equally: Yujie Guo, Tianlai Xia. ✉e-mail: r.oreilly@bham.ac.uk; mark.wallace@kcl.ac.uk

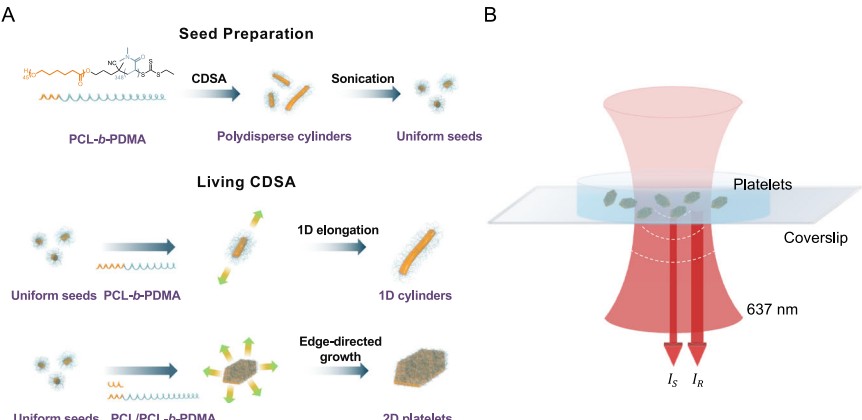

**Fig. 1 | Schematic illustration of living CDSA monitored by iSCAT. A** Uniform short micellar seeds were formed via sonication of polydisperse fibers of poly(ε-caprolactone)-*b*-poly(*N,N*-dimethylacrylamide), PCL-*b*-PDMA. 1D fibers and 2D platelets with controlled size and dispersity were then prepared via living CDSA by adding the block copolymer (PCL-*b*-PDMA) and unimer mixtures of PCL and PCL-*b*-PDMA to the seed solution, respectively. **B** iSCAT contrast results from interference between light scattered from growing platelets ($I_S$) and light reflected at the glass-liquid interface ($I_R$).

fiber[37]. Although providing easy access to nanoparticle kinetics, fluorescence microscopy's significant drawback is the requirement for extrinsic labels; given the relative size of a fluorophore to component monomers, it is perhaps unsurprising that impacts on the CDSA kinetics are reported[30]. Label-free techniques such as phase contrast or differential interference contrast microscopy can circumvent this issue but are typically limited in detection sensitivity[38–41].

To address the need for an in situ non-invasive method of characterizing living CDSA kinetics with high spatio-temporal single-particle resolution, here we apply interferometric scattering (iSCAT) microscopy to track CDSA in individual polymer particles. iSCAT is an intrinsically label-free technique capable of single-molecule resolution[42–44]. To date, iSCAT has mainly been used to study biological systems, including the mass measurement of individual proteins, the formation of lipid membranes, and biological diffusion using single-particle tracking of metal nanoparticles[45–47]. iSCAT has also been effectively utilized to monitor protein aggregation and actin polymerization processes[45]. Given the similarities shared between actin polymerization and 1D CDSA fiber formation, we reasoned that iSCAT might also be productively applied to the study of CDSA.

iSCAT relies on the interference between light scattered from an object of interest supported on a glass coverslip, and reference light reflected back from the same glass-sample interface (Fig. 1B). Compared to conventional interferometry, which is highly sensitive to environmental factors such as vibrations and temperature changes and often requires significant vibration isolation, the common path configuration of iSCAT allows the reference and scattered light to travel along the same optical path. This allows a simple and robust approach to measure changes in light scattering from small objects in a less controlled environment. For a diffraction-limited object, iSCAT signals appear as an Airy disc of concentric dark and light rings caused by the interference between the reflected and scattered signals. iSCAT offers lateral and axial resolutions of around 200–300 nm and 10–100 nm, respectively, and can further exploit optical super-resolution techniques to improve spatial precision[43,48]. In addition, iSCAT can operate at high temporal resolutions (up to 1 μs)[49] and monitor processes over long observation times without photobleaching.

CDSA kinetics are highly dependent on core chemistry[28], and to date, the majority of studies examining CDSA-mediated growth have focused on 1D fibers, with typically slow (hours to days) formation kinetics[30,36,50]. However, limited attention has been given to the study of 2D platelet kinetics, which are characterized by relatively rapid self-

assembly processes, distinct morphology, and ease of modification. Here, we chose poly(ε-caprolactone) (PCL)-based polymers as the core-forming block for the preparation of well-defined 1D fibers and 2D platelets. Compared with other commonly used core materials, the biocompatibility and biodegradability of PCL make it an attractive starting point for a range of materials for biomedical applications[11,51]. In addition, the crystallinity of the core-forming PCL block plays an essential role in determining the morphology and kinetics of CDSA particles: High crystallinity (i.e., high degree of polymerization) contributes to fast crystallization rates, commonly accompanied by self-nucleation and agglomeration; whereas core-forming blocks with lower crystallinity typically exhibit slower crystallization rates, resulting in irregular assembly shapes and a prolonged assembly process[52,53]. Here, $PCL_{45}$ was selected, as it provides a good balance between fast kinetics and tight morphological control. Using iSCAT, we mapped the kinetics and morphology of fiber and platelet growth from poly(ε-caprolactone)-*b*-poly(*N,N*-dimethylacrylamide) (PCL-*b*-PDMA) block copolymer and mixtures of PCL and PCL-*b*-PDMA, respectively.

## Results
### Real-time monitoring of living CDSA
To first summarize our overall method: Uniform seeds were prepared from polydisperse fibers of $PCL_{45}$-*b*-$PDMA_{348}$ formed in ethanol (Supplementary Fig. 3A), followed by sonication to produce short fibers with consistent size and length (mean length $24.6 \pm 4.9$ nm, Supplementary Figs. 3B, C). Seeds were then attached to the coverslip via spin-coating, with a silicone spacer placed on top to form a reaction chamber. This sample was then mounted on a custom-built microscope, ready for imaging (Supplementary Fig. 4)[54]. Unimer solutions were then introduced into the reaction chamber, followed by immediate iSCAT imaging to monitor the CDSA process (Fig. 1). An in-depth description of the experimental protocol and our instrumentation is provided in the "Methods" section and the Supplementary Information.

**1D fibers.** We initially applied iSCAT microscopy to monitor seeded-growth of 1D fibers: 50 μL of 2.51 nM (0.1 μg mL⁻¹) $PCL_{45}$-*b*-$PDMA_{348}$ seed solution was spin-coated onto the coverslip. $PCL_{73}$-*b*-$PDMA_{204}$ unimer was added into methanol to reach a final concentration of 0.06 μM (1.67 μg mL⁻¹) before being introduced into the reaction chamber and imaged using iSCAT.

Figure 2 A depicts a representative time series of iSCAT images collected during the formation of 1D fibers (637 nm, 4 μW μm⁻², 400 μs

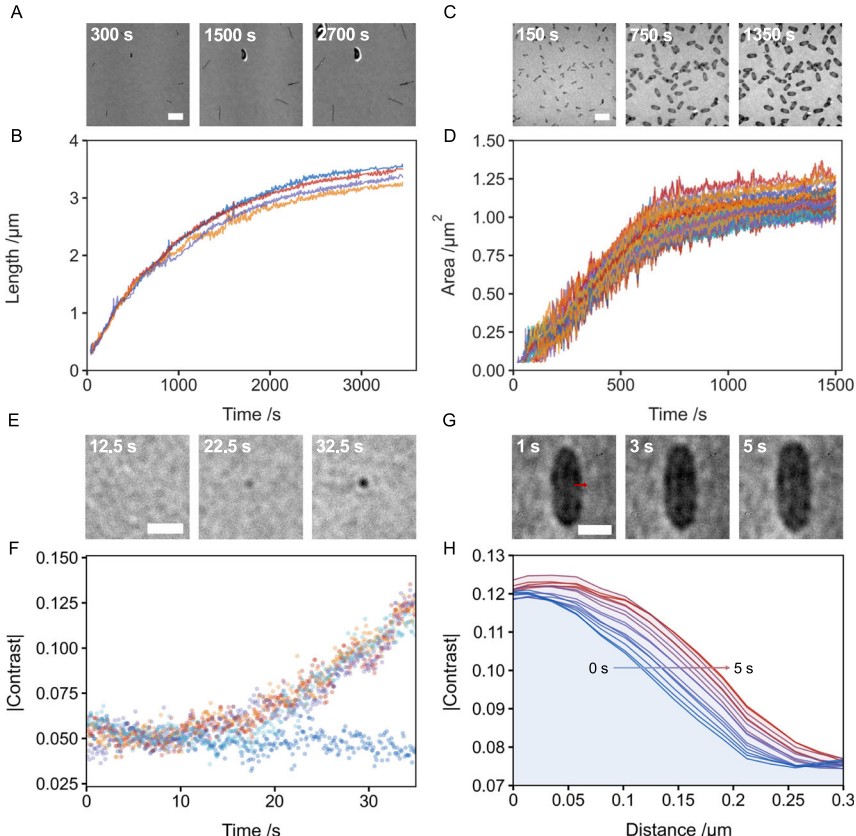

**Fig. 2 | iSCAT monitoring of living CDSA. A** iSCAT images of PCL$_{73}$-*b*-PDMA$_{204}$ 1D fiber growth (scale bar: 3 μm). **B** Length evolution of 1D fibers. **C** iSCAT images of PCL$_{45}$:PCL$_{45}$-*b*-PDMA$_{348}$ platelet growth (scale bar: 3 μm). **D** Size evolution of platelet area from 40 representative platelets taken from 3 repeats of the experimental conditions depicted in **C**. **E** iSCAT images of early diffraction-limited growth of a single platelet (scale bar: 1 μm). **F** Contrast evolution of individual platelets during the early stage. Blue markers represent the contrast evolution of the background, other colors indicate different individual platelets (100 ms time-lapse, 400 μs exposure). **G** iSCAT images of platelet growth at high speed (0.33 ms per frame, scale bar: 1 μm). **H** Contrast evolution of the cross section (indicated by the red arrow in **G**) at the platelet edge monitored at high imaging speed (contrast profiles were plotted every 33.33 ms with 100 frame averaging).

exposure time, 3 s$^{-1}$ per frame time-lapse). Upon unimer addition, uniform fiber elongation can be monitored as the reaction proceeds (the size of PCL$_{45}$-*b*-PDMA$_{348}$ seeds is below the detection limit of our setup, thus they are not observable; Supplementary Movie 1). Following image segmentation (see Supplementary Methods), the length evolution of individual fibers can be determined as shown in Fig. 2B. Applying a previously established kinetic model for 1D fiber growth[30], rate constants were extracted by fitting to the trajectories of length vs. time (Supplementary Methods section, Supplementary Fig. 5, 5.1 × 10$^{-3}$ s$^{-1}$). Comparing these data to previous work on poly(ferrocenyldimethylsilane)-*b*-polydimethylsiloxane (PFS$_{63}$-*b*-PDMS$_{513}$) 1D fibers (10–30 mg mL$^{-1}$ unimer and 0.1 mg mL$^{-1}$ seed), we observe faster kinetics in our PCL-based system (5.1 × 10$^{-3}$ s$^{-1}$ compared to 1.8 × 10$^{-4}$ to 2.13 × 10$^{-4}$ s$^{-1}$) despite lower unimer and seed concentration.

**2D platelets**. Following our initial imaging of 1D fiber formation, we next focused on 2D platelet growth, where the characteristic morphology and fast controllable assembly of platelets might be used to assess the capabilities of iSCAT for CDSA measurement. Seeds were deposited on glass coverslips at a surface density of ~0.16 μm$^{-2}$. Following the addition of 150 μL of 0.35 μM (3.33 μg mL$^{-1}$) PCL$_{45}$:PCL$_{45}$-*b*-PDMA$_{348}$ unimer mixtures, hexagonal platelets appear on the surface and uniformly grow in size as the reaction proceeds (Fig. 2C, Supplementary Movie 2; 637 nm, 4 μW μm$^{-2}$, 400 μs exposure, 1.5 s$^{-1}$ per frame time-lapse). We selected a PCL$_{45}$:PCL$_{45}$-*b*-PDMA$_{348}$ unimer mixture with the concentration of 0.35 μM (3.33 μg mL$^{-1}$) to ensure slow

growth of well isolated platelets. Keeping the seed surface density fixed, increasing the unimer concentration results in faster kinetics. Further details are provided in the "Methods" section below. Control experiments without the presence of seeds show no characteristic fast growth (Supplementary Fig. 6). We also explored the effect of changing the core chemistry: Platelets prepared with a poly($\eta$-octalactone) (POL)-based core-forming block exhibited less uniform morphology and size compared to the PCL-based system (Supplementary Fig. 7). Furthermore, we observed changes in platelet morphology with solvent (Supplementary Fig. 8), however, iSCAT was able to resolve CDSA assemblies under a wide range of solution conditions (Supplementary Fig. 9).

Image sequences were then segmented (Supplementary Methods) to extract parameters describing platelet morphology (area, long and short axis length, aspect ratio, and perimeter). Analysis of the time evolution of platelet area yields the kinetics of individual platelet growth (Fig. 2D). 210 platelets were recorded from 3 experimental repeats over a 25 min time-lapse. The individual growth trajectories of 40 representative platelets are shown in Fig. 2D, with their mean area evolution and standard deviation displayed in Supplementary Fig. 10. All platelets show similar growth kinetics and as shown in Supplementary Fig. 11A, at the final time point (1500 s), the platelet areas followed a Gaussian distribution with a mean of 1.12 μm$^2$ and a standard deviation of 0.077 μm$^2$. This relatively low standard deviation indicates that the majority of platelet areas converged toward a stable mean value over time. In addition, the area distributions of platelets collected at the end of in situ recording (Supplementary Fig. 11A) were

independent of the position of the platelet in the sample (Supplementary Fig. 11B). These data confirm the ability of living CDSA to produce uniform assemblies with controlled size and demonstrate that the in situ iSCAT imaging captures growth kinetics representative of the entire population.

During the early stages of CDSA, the platelet size is below the diffraction limit, and thus platelet area cannot be used to examine growth kinetics. However, the evolution of particle contrast can be used to report on this early stage (Fig. 2E, F, Supplementary Movie 3). As platelets are detected, individual spots begin to appear on the surface with their absolute contrast values increasing over time (becomes more negative, i.e., darker). Eventually, these diffraction-limited spots grow sufficiently that the characteristic shape becomes discernible (Fig. 2C). Similar to our measurements at later stages (Fig. 2D), during this early stage of growth, all platelets appear to increase in size at the same rate.

iSCAT imaging can also be used to monitor platelet growth at higher time resolution (Fig. 2G, H). To demonstrate this capability, we examined the progression of the position of a platelet edge, with contrast profiles extracted from platelet cross-sections as depicted in Fig. 2G. Data were collected with a temporal resolution of 333 µs (Supplementary Fig. 12, Supplementary Movie 4). A plot of the contrast profile evolution at 33.3 ms intervals with 100-frame averaging is shown in Fig. 2H.

## Comparison of characterization methods

With iSCAT established as a method to image CDSA assemblies, we sought to assess the reliability and accuracy of iSCAT for CDSA characterization in comparison to conventional methods for examining platelet morphology and size.

We compared platelet parameters extracted from iSCAT imaging with those obtained from identical samples using AFM, TEM, and CLSM. Assembly of fluorescently-labeled platelets was initiated by adding 10 µL of 1 mM (10 mg mL$^{-1}$) aminochloromaleimide (ACM)[55]-coupled homopolymer and block copolymer unimer mixtures (ACM-PCL$_{45}$:ACM-PCL$_{45}$-$b$-PDMA$_{348}$, 1:1, w:w) into 1 mL 0.25 µM (10 µg mL$^{-1}$) PCL$_{45}$-$b$-PDMA$_{348}$ seed solution in ethanol, followed by mixing via shaking. At predetermined time points, identical aliquots were taken, quenched by water addition, and analyzed using each technique. Further details for AFM, TEM, and CLSM measurement are provided in the "Methods" section.

Figure 3 shows that platelet morphology and area information extracted from AFM, TEM, CLSM, and iSCAT are comparable (Supplementary Fig. 13). Under these reaction conditions, platelet formation is rapid, with the assembly completed within 2 min of our first time point; essentially faster than the time resolution of AFM, TEM, and CLSM measurements.

## Platelet growth kinetics

To characterize the kinetics of living CDSA, we adapted a kinetic model recently developed by Gao and co-workers[56], where edge-directed 2D platelet kinetics is described by

$$\frac{dN_t}{dt} = -k[4\pi B n_a^2 (N_0 - N_t)]^{\frac{1}{2}} N_t,\qquad(1)$$

where $N_t$ is the number of unimers remaining in the reaction at time $t$; $k$ is the overall growth rate constant; $n_a$ is the number of unimers in the exposed areas per unit perimeter; $N_0$ is the initial number of unimers; and $B$ is the area contributed per unimer, calculated from the ratio of final platelet area to initial number of unimers (Supplementary Equation (6)). The area per unimer ($B$) is considered as a constant, linking the experimentally determined platelet area to the consumption of unimers[56]. Consequently, the evolution of platelet area ($A_t$) with time ($t$) can be expressed simply in terms of an effective overall rate

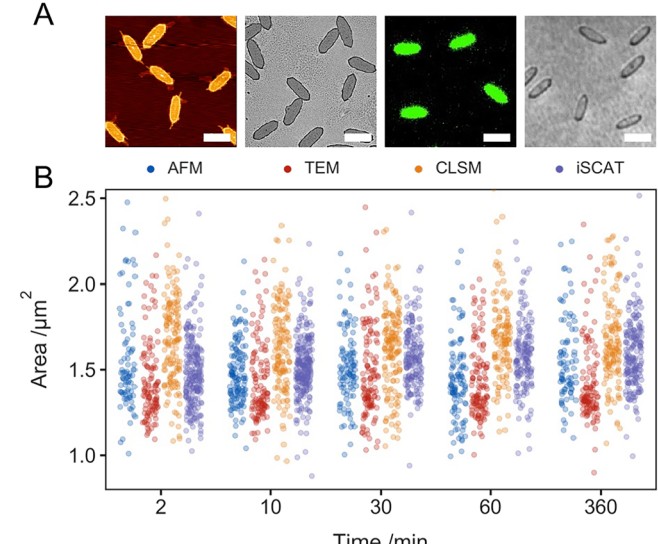

**Fig. 3 | Comparison of 2D platelet characterization methods. A** Images collected using AFM, TEM, CLSM, and iSCAT of a sample of ACM-PCL$_{45}$:ACM-PCL$_{45}$-$b$-PDMA$_{348}$ platelets collected after 6 h of growth (scale bars: 2 µm). **B** Comparison of ACM-PCL$_{45}$:ACM-PCL$_{45}$-$b$-PDMA$_{348}$ platelet area distributions over time measured with AFM (blue), TEM (red), CLSM (orange), and iSCAT (purple). For TEM measurements, platelets were deposited onto copper grids without staining.

constant, $k'$ (see Supplementary Methods):

$$A_t = \frac{BN_0}{N_{seed}}\left(1 - \frac{2}{\cosh(k't)+1}\right).\qquad(2)$$

We first validated our method by ensuring that the time-dependence of platelet morphology sampled from a CDSA reaction at predetermined intervals and then analyzed by both iSCAT and AFM was consistent (Supplementary Fig. 14).

To quantify the kinetics of platelet growth, we then examined the dependence of platelet size and morphology on unimer concentration, seed concentration, and solvent conditions (Fig. 4). Platelets were prepared in bulk under various reaction conditions, with sample aliquots removed at predefined time points to fully capture the reaction kinetics. These samples were then characterized by iSCAT (2 µW µm$^{-2}$, 637 nm, 900 µs exposure time; see further details in the "Methods" section). A minimum of 100 platelets were analyzed at each time point for each condition.

As expected, a higher initial unimer concentration leads to faster platelet growth, producing platelets with larger final surface area (Fig. 4A). A linear dependence of the overall rate constant on unimer concentration was observed with an overall reaction order of 0.41 (Supplementary Fig. 15A). In addition, the coefficient of determination ($R^2$) reported for all concentrations indicates that the model provides a good fit. This deviation from simple first-order kinetics has been reported previously and attributed to conformational effects on the block copolymer during assembly[30,32]. We also examined the dependence of platelet formation on seed concentration (0.25–1.13 nM), keeping unimer concentration and solvent conditions fixed (Fig. 4B). As expected, a higher seed concentration produces platelets with smaller final sizes.

The impact of tetrahydrofuran (THF), a good solvent for the crystallizable core-forming block (PCL), was investigated. Figure 4C illustrates the overall inhibitory effect of THF addition: Increasing the volume fraction of good solvent hinders the platelet growth rate, as crystallization is slowed due to enhanced polymer solubility[30]. Generally, we observe that the final area of platelets generated from fixed unimer and seed concentrations remains consistent, regardless of solvent compositions (0–3% of THF). However, notably, this does not

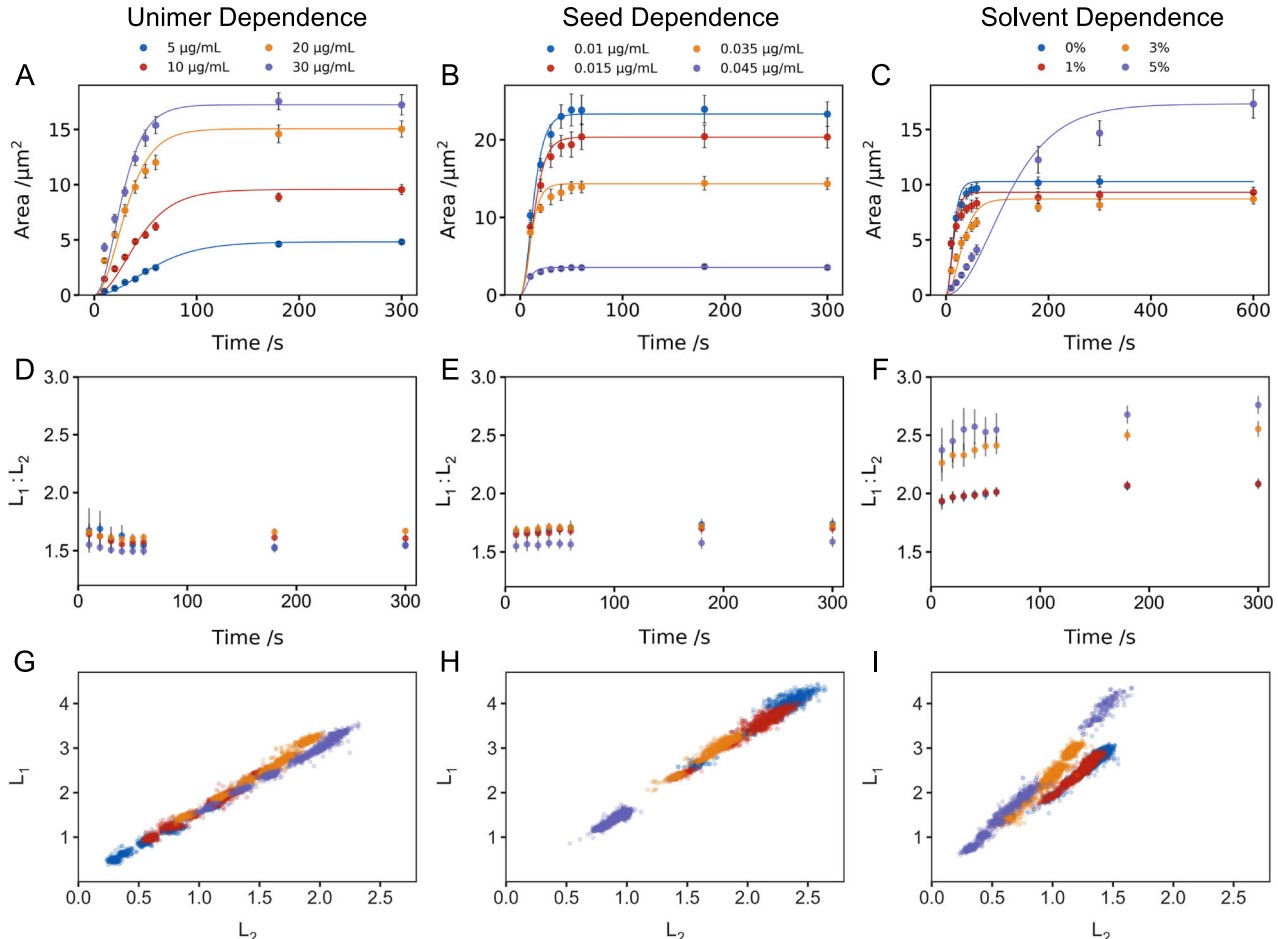

**Fig. 4 | Dependence of platelet kinetics and morphology on reaction conditions. A** *Unimer dependence.* Unimer ($PCL_{45}$:$PCL_{45}$-*b*-$PDMA_{348}$, 1:1, w:w) concentration was varied from 0.53, 1.05, 2.10 to 3.15 µM (5, 10, 20 to 30 µg mL$^{-1}$). Seed concentration was fixed at 0.13 nM (0.005 µg mL$^{-1}$). The effective rate constants ($k'$) extracted from the fitting for each unimer concentration (from low to high) are 0.032, 0.042, 0.058, and 0.066 s$^{-1}$, respectively. The $R^2$ values for the fits at each concentration (from low to high) are as follows: 0.9919, 0.9437, 0.9331 and 0.9429, respectively. **B** *Seed dependence.* Seed concentration was varied from 0.25, 0.38, 0.88 to 1.13 nM (0.01, 0.015, 0.035 to 0.045 µg mL$^{-1}$). Unimer concentration was fixed at 6.31 µM (60 µg mL$^{-1}$). The effective rate constants ($k'$) extracted from the fitting for each seed concentration (from low to high) are 0.139, 0.131, 0.170, and 0.218 s$^{-1}$, respectively. The $R^2$ values for the fits at each concentration (from low to high) are as follows: 0.9529, 0.9366, 0.7911 and 0.7983, respectively. **C** *Solvent* *dependence.* THF volume fraction was varied from 0, 1, 3 to 5%. Unimer and seed concentrations were fixed at 2.10 µM and 0.38 nM (20 and 0.015 µg mL$^{-1}$), respectively. The effective rate constants ($k'$) extracted from the fitting for each THF concentration (from low to high) are 0.124, 0.123, 0.057, and 0.015 s$^{-1}$, respectively. The $R^2$ values for the fits at each concentration (from low to high) are as follows: 0.7942, 0.4762, 0.8333 and 0.9703, respectively. Error bars in **A**, **B**, **C** represent the standard deviations of the platelet area distribution, calculated from 100 platelets ($n = 100$) for each time point. **D**, **E**, **F** are the temporal evolution of the ratio between long ($L_1$) and short axis ($L_2$) ($L_1$: $L_2$, aspect ratio). Error bars in **D**, **E**, **F** represent the standard deviations of the platelet aspect ratio distribution, calculated from 100 platelets ($n = 100$) for each time point. **G**, **H**, **I** are the distribution of the length of the long ($L_1$) and short axis ($L_2$) for the 100 individual platelets ($n = 100$) analyzed in **A**–**C**.

hold true for the highest THF concentration we explored (Fig. 4C, purple, 5%). Similar results have been observed previously, attributing the increased final size to the solubilization of the seeds themselves at high THF content. This solubilization leads to a decrease in crystalline nuclei, thereby providing an additional source of unimers for remaining nuclei[15].

Alongside measurements of growth kinetics based on platelet area, the impact of unimer, seed, and solvent variation on the platelet morphology was also examined. Figures 4D–F show the time evolution of the platelet aspect ratio ($L_1$: $L_2$). Figures 4G–I show the underlying changes in long ($L_1$) and short ($L_2$) axis lengths with experimental conditions. Both unimer and seed concentrations had little impact on the platelet shape during assembly. Conversely, alterations in the THF content in the system (from 0 to 5%) significantly impacted platelet shape. While the platelet aspect ratio ($L_1$: $L_2$) remained relatively constant during platelet formation for a particular THF concentration (Fig. 4F), an increased THF content led to a more pronounced preference for unimer addition along the longer axis ($L_1$), resulting in the

formation of elongated platelets as illustrated in Fig. 4I and Supplementary Fig. 16. However, the mechanism leading to this asymmetry remains unknown and warrants further study.

Overall, for all conditions tested (unimer concentration, seed concentration, and solubility), our simple kinetic model (Eq. (2)) for edge-directed growth provides a good description of our experimental observations. It can also be noted that the core chemistry significantly affects the kinetics. Compared with 1D fiber formation using a poly(-ferrocenyldimethylsilane)-based core-forming blocks[30] or 2D nanosheet formation from poly(cyclopentenylene vinylene)-based core-forming blocks[32], our PCL-based system exhibits faster kinetics in both 1D and 2D assembly.

## Sequential compositional control via edge-directed platelet growth

iSCAT is highly sensitive to changes in object thickness and refractive index. This can be exploited to provide additional information on nanoscopic assembly/disassembly processes. For instance, changes in

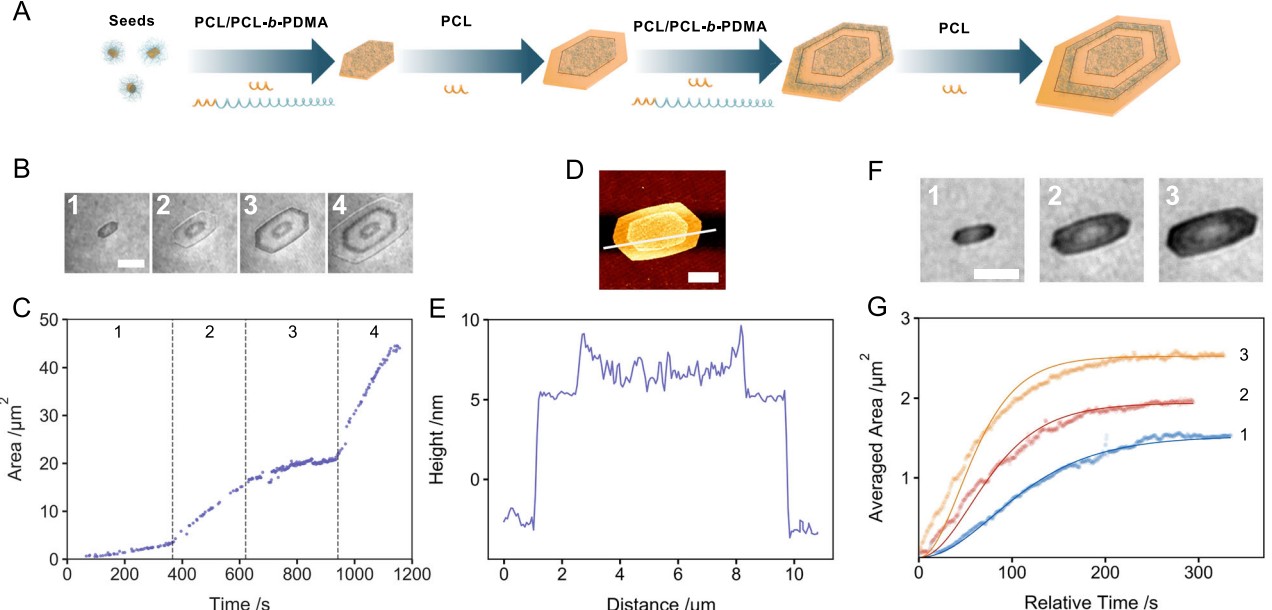

**Fig. 5 | Multi-annulus platelet formation kinetics. A** Schematic illustration of 4-annulus platelet formation using varied unimer composition via living CDSA. **B** Montage of iSCAT images during 4-annulus platelet growth (scale bar: 3 µm). 50 µL of 2.51 nM (0.1 µg mL⁻¹) seed solution was spin-coated onto the surface twice, and 4-annulus platelets were formed via alternating addition of PCL$_{45}$:PCL$_{45}$-$b$-PDMA$_{348}$ and PCL$_{45}$ unimer solutions at concentrations of 0.18, 0.39, 0.35 and 0.77 µM (1.67, 2.08, 3.33 and 4.17 µg mL⁻¹) for each respective annulus. **C** Size evolution of a 4-annulus platelet. **D** AFM image of a 2-annulus platelet (scale bar: 3 µm). 50 µL of 2.51 nM (0.1 µg mL⁻¹) seed solution was spin-coated onto the surface, 0.18 µM (1.67 µg mL⁻¹) PCL$_{45}$:PCL$_{45}$-$b$-PDMA$_{348}$ mixtures (PCL concentration: 0.15 µM, 0.83 µg mL⁻¹) in methanol was added to form the first annulus. 0.15 µM (0.83 µg mL⁻¹) PCL$_{45}$ was then added. **E** Height profiles corresponding to the cross-section drawn in **D**. **F** Montage of iSCAT images during 3-annulus platelet growth (scale bar: 2 µm). 50 µL of 2.51 nM (1.67 µg mL⁻¹) seed solution was spin-coated onto the surface, the PCL$_{45}$:PCL$_{45}$-$b$-PDMA$_{348}$ unimer mixtures were added at concentrations of 0.44, 0.59, and 0.87 µM (4.2, 5.6 and 8.3 µg mL⁻¹) for each annulus. **G** Mean size evolution of 3-annulus platelets. Rate constants extracted from the fitting for each annulus are 0.017, 0.024, and 0.030 s⁻¹, respectively.

scattering properties have previously been utilized to study 2D lipid bilayer formation and phase transitions[46,57]. Following CDSA, the platelet perimeter remains active for the templated edge-directed growth of additional platelet area[26]. Therefore, multi-annulus platelets can be prepared by sequential unimer addition. The growth occurs radially from the edges of the platelets, with each successive annulus extending the preceding one. By varying the unimer composition, the composition/thickness of each annulus can be adjusted, which allows us to utilize the contrast mechanism of iSCAT to investigate the kinetics and properties of individual annuli during platelet assembly.

4-annulus platelets were prepared by sequentially adding mixtures of PCL$_{45}$:PCL$_{45}$-$b$-PDMA$_{348}$ (1:1, w:w) and pure PCL$_{45}$ (Fig. 5A, Supplementary Movie 5, further details in "Methods" section). Figure 5B shows that platelet annuli formed from PCL$_{45}$:PCL$_{45}$-$b$-PDMA$_{348}$ mixtures exhibit a higher (more negative) iSCAT contrast (i.e., darker) compared to annuli prepared from pure PCL$_{45}$. This difference is likely due to a combination of both refractive index and thickness variations resulting from differences in polymer composition. As iSCAT is highly sensitive to variations in refractive index, with a detection limit as low as 0.005[43,58], the refractive index difference between PCL ($n = 1.476$) and PDMA ($n = 1.430–1.456$)[59,60] can contribute to the observed contrast variation. Additionally, the increased thickness of platelet annuli formed from PCL$_{45}$:PCL$_{45}$-$b$-PDMA$_{348}$, compared to pure PCL$_{45}$, as shown in the AFM measurements of two-annulus platelets prepared following the same protocol (Figures 5D, E and Supplementary Fig. 17), can also lead to a path length difference between the scattered and reference signals. However, despite AFM providing direct evidence that thickness variations contribute significantly to the observed iSCAT contrast, it remains challenging to clearly decouple the impact of refractive index and thickness, as both factors influence the optical path length at the nanoscale.

Figure 5 C illustrates the size evolution for the platelet shown in Fig. 5B. Growth kinetics are dependent on platelet composition, with the first and third annuli exhibiting different kinetics as compared to the second and fourth annuli. To explore this observation further, we prepared 3-annulus platelets where the unimer composition was fixed, but the concentration of PCL$_{45}$:PCL$_{45}$-$b$-PDMA$_{348}$ was varied for each annulus (Fig. 5F, Supplementary Movie 6, further details in "Methods" section). As shown in Fig. 5F, with no expected obvious difference in annulus thickness, the boundaries between each annulus are less distinct, albeit still identifiable (to clarify the boundaries, each annulus was color-coded as shown in Supplementary Fig. 18A). The outermost boundary within each annulus appears discernibly darker (with a larger negative contrast) compared to the center. We speculate that this effect is potentially caused by a nonuniform distribution of PCL$_{45}$ and PCL$_{45}$-$b$-PDMA$_{348}$ within each annulus due to differences in their crystallization rates[61,62]. However, although providing some contrast variations, this effect was insufficient to render significant changes in growth kinetics. Figure 5G illustrates the progression of averaged platelet area (calculated from averaging three 3-annulus platelets displayed in Supplementary Fig. 18B), with each successive unimer addition synchronized to the corresponding time of addition. The rate constants extracted for each annulus were essentially unchanged from those in Fig. 4A. As expected, a higher unimer concentration resulted in a faster assembly rate. As a result, besides the dimensional and morphological information, iSCAT can also probe into the nanoscopic surface information of individual platelets based on their contrast variations.

## Discussion
Taken together, our results demonstrate the potential of using iSCAT microscopy as a label-free tool for the in situ reporting the morphological and dimensional information of CDSA assemblies, ranging from

1D fiber formation to 3D thickness information of multi-annulus structures. With its excellent sensitivity and sub-millisecond temporal resolution, iSCAT successfully provides detailed insights into the CDSA process, including early-stage growth and high-resolution imaging of the growth dynamics.

This method has enabled us to quantify the dependence of living CDSA kinetics on unimer concentration, seed concentration, and selective solubility. Living CDSA shares obvious similarities with living covalent polymerization, where unimer and exposed active sites at the platelet edges are analogous to the monomer and initiator in the reaction. Here, 2D platelet formation of PCL$_{45}$:PCL$_{45}$-*b*-PDMA$_{348}$ was well described by a kinetic model based on such assembly at the living edge of the forming platelet, with a sub-unitary overall order of reaction and dependence of growth kinetics on the nature of the block copolymer[32,56]. This deviation from simple first-order kinetics has been previously attributed to conformational dispersity of the block copolymer[28,30]. This is particularly intriguing when considering the variation in contrast within growth annuli in our examination of the growth of multi-annulus platelets (Fig. 5). Secondly, observed variations in contrast across individual annuli are of note (Fig. 5F). One interpretation could be a change in composition, perhaps caused by a variation in the relative rates of deposition of PCL$_{45}$ and PCL$_{45}$-*b*-PDMA$_{348}$. These observations warrant further investigation, mapping how different building units might assemble differently as 2D platelet shape evolves. Lastly, in a multi-annulus platelet, the contrast variations between each annulus can be attributed to changes in both refractive index and thickness resulting from variations in composition. The interplay between these two factors is challenging to fully decouple without additional information. iSCAT's ability to provide rapid, label-free insights into structures and composition suggests significant potential for detecting relative changes in polymeric particle properties. We anticipate that further development of methods for decoupling thickness and refractive index effects will enhance the quantitative capabilities of iSCAT, leading to broader applications.

Beyond its capabilities for in situ monitoring, iSCAT also holds much promise as a tool for ex situ characterization of particle morphology due to its fast data collection, low cost, simple sample preparation, and high throughput. Additionally, this technique can be implemented using simple commercially available optical components compatible with fluorescence microscopy[48,63]. However, iSCAT is not without limitations: Microscopy limits the field of view in any single image ($20 \times 20\,\mu m$ in our setup), requiring parallelization or surface scanning to monitor the growth of larger numbers of particles simultaneously. iSCAT also requires a reference light-field and thus is limited to studies at or near a surface. We note that here we saw no difference in kinetics between platelets grown in solution and those growing in situ at the coverslip surface. However, the presence of this surface is inevitably different at some level from growth conditions in the bulk solution. Nevertheless, confocal variants of iSCAT hold the potential for 3D imaging much deeper into the sample solution[44,48]. Here we have focused on 2D CDSA, but the interferometric nature of iSCAT also embeds axial information; enabling, for example, the measurement of nanoparticle axial position and size[47,48,64–66]. As illustrated in Supplementary Fig. 19, we applied recent methods that exploit defocused image stacks to generate quantitative phase information[67]. However, to establish an accurate relationship between the sample thickness and the corresponding phase information, calibration is required, using samples with controlled refractive indices and varied thicknesses. We envisage iSCAT has great potential to be applied to profiling more complex CDSA processes, with three-dimensional shape at high spatiotemporal resolution[68–70]. However, it is important to emphasize that in terms of the scope of our current study, which mainly focuses on the two-dimensional CDSA and growth, limited information on the 3D height variation in our 2D system can be extracted. In addition, there remains considerable scope to further enhance our detection sensitivity; for example, by applying recent reports of machine learning[71] and optimization of reflected vs. scattered signals[72] to improve iSCAT mass sensitivity.

Overall, iSCAT is a potent technique that offers a different perspective on improving CDSA-based nanoparticle engineering. Here, we have demonstrated the applicability of iSCAT to study PCL-based 1D fibers and 2D platelets. Real-time label-free single platelet imaging provides a rich source of information hitherto inaccessible by most commonly used techniques. This combination of speed, resolution, and simplicity further places iSCAT as a valuable tool for the design and characterization of complex polymer particles generated by CDSA.

## Methods
### Materials
Unless otherwise stated, all chemicals and solvents were used as received without further purification. Sodium ethanethiolate (90%) and borane tetrahydrofuran complex solution (1.0 M in THF) were purchased from Alfa Aesar. Carbon disulfide (≥99%), solid iodine (≥99%) and 4,4'-azobis(4-cyanovaleric acid) (ACVA, 98%) were purchased from Merck.

The synthesis of dual head chain transfer agent (CTA) of 2-cyano-5-hydroxypentan-2-yl ethyl carbonotrithioate (CHPET) follows previous reports[11]. CTA and diphenylphosphate (DPP, 99%, Merck) were dried over P$_2$O$_5$ in a desiccator under static vacuum for 1 week before use. $\varepsilon$-caprolactone (99%, ACROS Organics) was vacuum distilled twice over CaH$_2$ before being introduced in the glove box and used. 2,2'-azobis(2-methylpropionitrile) (AIBN, 98%, Merck) was recrystallized twice from methanol and stored at 4 °C in the dark. 1,4-dioxane (anhydrous, 99.8%) and *N, N*-dimethylacrylamide (DMA, 99%, contains 500 ppm monomethyl ether hydroquinone as inhibitor) were purchased from Merck. DMA was passed through a basic alumina plug to remove the inhibitor before use. Dry solvents were used directly from a drying and degassing inert solvent tower system. Amino-chloromaleimide (ACM) fluorescent dye was synthesized following the method provided below[55].

### Polymer synthesis and characterization
**Synthesis of PCL$_{45}$.** In a nitrogen-filled glove box (oxygen and water content lower than 0.1 ppm), solutions of diphenylphosphate (DPP, 17.0 mg, 1 eq) in dry toluene (2.5 mL) and dual-head CTA (17.0 mg, 1 eq) in dry toluene (1 mL) were added to $\varepsilon$-caprolactone dry toluene solution (543.3 mg, 70 eq in 1.5 mL). After stirring for 6 h at room temperature, the solution was removed from the glove box, precipitated three times into cold diethyl ether dropwise, and collected by centrifugation. $^1$H NMR (400 MHz, CDCl$_3$): $\delta$ (ppm): 4.10 (t, 2H, C**H**$_2$OCO), 4.04 (t, 90H, CH$_2$C**H**$_2$O), 3.62 (m, 2H, C(CN)C**H**$_2$CH$_2$), 3.32 (q, 2H, SC**H**$_2$CH$_3$), 2.28 (90H, OCOC**H**$_2$CH$_2$), 1.86 (s, 3H, C(CN)(C**H**$_3$)CH$_2$), 1.72–1.54 (180H, OCOCH$_2$C**H**$_2$CH$_2$**H**$_2$), 1.45-1.28 (90H, OCOCH$_2$CH$_2$C**H**$_2$CH$_2$). SEC (Chloroform, PMMA standard): $M_n$ = 12.9 kg mol$^{-1}$, $Đ_M$ = 1.05.

To synthesize ACM, 2,3-dichloromaleic anhydride (1 eq) and 4-aminobutanoic acid (1 eq) were dissolved in 10 mL acetic acid (AcOH) in a round-bottom flask. The reaction mixture was heated at 150 °C for 6 h under constant stirring. Upon completion, the mixture was concentrated under reduced pressure to remove the AcOH. The crude residue was dissolved in 5 mL dichloromethane (DCM) and purified via flash column chromatography using 5% AcOH in DCM as the eluent. The product was isolated as an orange solid. Then the above product was dissolved in 20 mL THF in a reaction flask. Sodium carbonate (2.5 eq) and butylamine (1.05–1.1 eq) were added, and the reaction mixture was stirred at room temperature. The reaction progress was monitored by thin layer chromatography, confirming the consumption of the above product within 30 min to 4 h. The solvent was evaporated under reduced pressure, and the residue was dissolved in

150 mL DCM. The organic layer was washed with water (2 × 150 mL), dried over anhydrous sodium sulfate, and filtered. The filtrate was concentrated under reduced pressure, and the product was purified via column chromatography on silica gel using petroleum ether/ethyl acetate as the eluent. The product was obtained as a yellow-orange solid.

To synthesize ACM-labeled $PCL_{45}$, ACM was coupled to the $PCL_{45}$ polymer backbone by esterification. In a typical coupling reaction, $PCL_{45}$ (100.0 mg, 1 eq), ACM (16.1 mg, 3 eq), 4-dimethylaminopyridine (DMAP) (2.3 mg, 1 eq) and $N,N'$-dicyclohexylcarbodiimide (DCC) (38.3 mg, 10 eq) were mixed together in an ampule with 2 mL $CHCl_3$. The solution was left stirring at room temperature for 2 days. The solution was then filtered with the filtrate precipitated in diethyl ether three times and the resultant polymer was dried under vacuum.

**Synthesis of $PCL_{45}$-$b$-$PDMA_{348}$.** $PCL_{45}$ (100.0 mg, 1 eq), DMA (736.3 mg, 400 eq), and AIBN (0.3 mg, 0.1 eq) were dissolved in 1 mL 1,4-dioxane and placed in an ampule. The solution was then freeze-pump-thawed three times and heated for 2 h at 70 °C. The reaction was quenched by immersion of the ampule in liquid nitrogen and the polymer was precipitated in ice-cold diethyl ether three times before being dried under vacuum and analyzed. SEC (chloroform, PMMA standard): $M_n = 46.3$ kg mol$^{-1}$, $Đ_M = 1.24$.

To synthesize ACM-labeled $PCL_{45}$-$b$-$PDMA_{348}$, ACM was coupled to the $PCL_{45}$-$b$-$PDMA_{348}$ polymer backbone by esterification. In a typical coupling reaction, $PCL_{45}$-$b$-$PDMA_{348}$ (200 mg, 1 eq), ACM (4.3 mg, 3 eq), DMAP (0.6 mg, 1 eq), and DCC (10.3 mg, 10 eq) were mixed in an ampule with 2 mL $CHCl_3$. The solution was left stirring at room temperature for 2 days. The solution was then filtered, and the filtrate precipitated in diethyl ether three times and the resultant polymer was dried under vacuum.

**NMR characterization.** All $^1$H NMR spectra were recorded on a Bruker 400 MHz (DPX-400) spectrometer. Chemical shifts ($\delta$) are reported in parts per million (ppm) relative to internal standard tetramethylsilane at $\delta = 0$. Samples were prepared in deuterated chloroform ($CDCl_3$) and referenced to residual non-deuterated signal of solvent ($CDCl_3$ at 7.26 ppm $^1$H-NMR). The resonance multiplicities are described as s (singlet), d (doublet), t (triplet), q (quartet), or m (multiplet).

**SEC characterization.** SEC analysis was performed on an Agilent 1260 Infinity II system combined with refractive index (RI) and ultraviolet (UV) detectors ($\lambda = 309$ and 360 nm), equipped with a PLGel 3 μm (50 × 7.5 mm) guard column and two PLGel 5 μm (300 × 7.5 mm) mixed-D columns, mobile phase (eluent) is $CHCl_3$ with 0.5% triethylamine (TEA). Molecular weight and molecular weight distributions were calibrated against poly(methyl methacrylate) (PMMA) standards and analyzed using Agilent SEC software.

**CDSA preparation**

**Preparation of $PCL_{45}$-$b$-$PDMA_{348}$ seed solution.** 20 mg $PCL_{45}$-$b$-$PDMA_{348}$ was added into 4 mL ethanol. The mixtures were heated at 70 °C without stirring on a heating block for 3 h before cooling to room temperature and then aging for 5 days to yield micron-length polydisperse fibers (Supplementary Fig. 3A). The crystalline fibers were then sonicated using a Bandelin Sonopuls sonication probe in a dry-ice/acetone bath for 20 min. The polymer solution was exposed to 60 cycles of 20 s bath sonication with an interval 100 s to yield a short crystalline seed stock solution (Supplementary Fig. 3B). For determination of the average length of the seed, a minimum number of 100 seeds were analyzed with TEM (Supplementary Fig. 3C).

**Bulk preparation of $PCL_{45}$:$PCL_{45}$-$b$-$PDMA_{348}$ platelets.** 10 mg mL$^{-1}$ unimer stock solution ($PCL_{45}$:$PCL_{45}$-$b$-$PDMA_{348}$ or ACM-$PCL_{45}$:ACM-

$PCL_{45}$-$b$-$PDMA_{348}$) was prepared by mixing $PCL_{45}$ (or ACM-$PCL_{45}$) with $PCL_{45}$-$b$-$PDMA_{348}$ (or ACM-$PCL_{45}$-$b$-$PDMA_{348}$) at a 1:1 weight ratio in a good solvent (THF or Chloroform). A working seed solution was prepared by diluting the stock seed solution in ethanol to form a targeted concentration, then aged for at least 1 h. Then a small volume of unimer stock solution was added to a dispersion of seed solution in a screw cap vial followed by vigorously shaking by hand for 5 s, the final unimer concentration was varied by adjusting the volume of stock solutions added to each reaction. At predetermined time points, 140 μL sample mixture was withdrawn and added to 140 μL of deionized water to quench the reaction, followed by subsequent analysis.

**In situ 1D fiber growth.** 50 μL of 2.51 nM (0.1 μg mL$^{-1}$) $PCL_{45}$-$b$-$PDMA_{348}$ seed solution was spin-coated onto the cleaned coverslip twice at 3200 rpm for 50 s, immediately followed by 4000 rpm for 30 s. A cleaned silicone spacer was then placed onto the seed-coated coverslip to form a reaction chamber. This reaction chamber has then been mounted above the objective with the objective being adjusted to focus on the surface (Supplementary Fig. 4). $PCL_{73}$-$b$-$PDMA_{204}$ unimer stock solutions were diluted in methanol to achieve the targeted concentration (freshly prepared every time, 0.06 μM, 1.67 μg mL$^{-1}$) and added into the reaction chamber, followed by iSCAT imaging immediately. A laser power density of 4 μW μm$^{-2}$ at 637 nm, a camera exposure time of 400 μs, and an overall time-lapsed frame rate of 3 s$^{-1}$ were selected.

**In situ 2D platelet growth.** To obtain data shown in Fig. 2C, D, plasma-cleaned coverslips were placed in 50 mL 0.002 μg mL$^{-1}$ poly($\varepsilon$-caprolactone)-$b$-poly(2-vinylpyridine) ($PCL_{45}$-$b$-$P2VP_{134}$) seed ethanol solution for 36 h to achieve 2D surface seed density of ~0.16 μm$^{-2}$. Before the reaction, the coverslip was gently rinsed with ethanol and dried under a stream of nitrogen. 150 μL of 0.35 μM (3.33 μg mL$^{-1}$) $PCL_{45}$:$PCL_{45}$-$b$-$PDMA_{348}$ unimer mixtures were then added onto the seed-modified surface followed by recording immediately (637 nm, 4 μW μm$^{-2}$, 400 μs exposure time, 1.5 s$^{-1}$ per frame time-lapse).

For other experiments, spin coating was used to prepare seeds-coated surface: 50 μL of 2.51 nM (0.1 μg mL$^{-1}$) $PCL_{45}$-$b$-$PDMA_{348}$ seed solution was spin-coated onto the cleaned coverslip twice at 3200 rpm for 50 s, immediately followed by 4000 rpm for 30 s. A reaction chamber was constructed following the same protocol introduced above, $PCL_{45}$:$PCL_{45}$-$b$-$PDMA_{348}$ unimer stock solutions were diluted in methanol to achieve the targeted concentration and added into the reaction chamber, followed by iSCAT imaging immediately.

For data shown in Fig. 2E, F, 150 μL of 0.35 μM (3.33 μg mL$^{-1}$) $PCL_{45}$:$PCL_{45}$-$b$-$PDMA_{348}$ unimer mixtures were added onto the surface. A laser power density of 4 μW μm$^{-2}$ at 637 nm, a camera exposure time of 400 μs, and an overall time-lapsed frame rate 10 s$^{-1}$ were selected. For data shown in Fig. 2G, H, 150 μL of 0.29 μM (2.78 μg mL$^{-1}$) $PCL_{45}$:$PCL_{45}$-$b$-$PDMA_{348}$ unimer mixtures were added onto the surface. A laser power density of 24 μW μm$^{-2}$ at 637 nm, a camera exposure time of 80 μs, and an overall time-lapsed frame rate of 3000 s$^{-1}$ were selected.

Multi-annulus platelets were prepared by alternately adding $PCL_{45}$:$PCL_{45}$-$b$-$PDMA_{348}$ mixtures and $PCL_{45}$ alone. $PCL_{45}$:$PCL_{45}$-$b$-$PDMA_{348}$ mixtures were first added onto a seed-coated surface to form the first annulus. After ~5 min, the reaction solution was removed from the chamber and the chamber was rinsed gently with methanol to remove unimer residues, followed by the addition of $PCL_{45}$ solution. This alternating sequence was repeated until the desired number of annuli was achieved. Recording started immediately after each unimer addition. A laser power density of 4 μW μm$^{-2}$ at 637 nm, a camera exposure time of 400 μs, and an overall time-lapsed frame rate of 1 s$^{-1}$ were selected. For the experiment in Fig. 5F, $PCL_{45}$:$PCL_{45}$-$b$-$PDMA_{348}$ mixtures with sequentially increasing unimer concentration were added at each annulus. A laser power density of 4 μW μm$^{-2}$ at 637 nm, a camera exposure time of 400 μs, and an overall time-lapsed frame rate 0.5 s$^{-1}$ were selected.

## Comparison of analysis methods

**TEM characterization.** TEM imaging was performed on a JEOL 1400 Bio (1720/GB06) microscope at an acceleration voltage of 80 kV. CDSA samples were deposited onto copper grids without staining and dried naturally before analysis. For the determination of the average size of the platelets, a minimum number of 100 platelets were analyzed.

**AFM characterization.** AFM samples were prepared by drop-casting and drying 10 μL of assemblies in ethanol onto mica attached to a Sigmacote-treated silicon wafer. Imaging and analysis were performed on a JPK Nanowizard 4 in quantitative imaging (QI) mode. AFM tips (PPP-NCHAuD) were purchased from NANOSENSORSTM, with a resonance frequency window of 204–497 kHz and a force constant of 10–130 nN. For the determination of the average size of the platelets, a minimum number of 100 platelets were analyzed.

**CLSM characterization.** An inverted FV3000 (Olympus) confocal microscope with 20× and 60× oil immersion objectives was used for imaging. Scan rates of 1 μs pixel$^{-1}$ at 512 by 512 pixels to 1024 by 1024 pixels were used. For platelet characterization, a minimum number of 100 platelets were segmented manually using Fiji image analysis software[73].

**iSCAT characterization.** A wide-field iSCAT was previously constructed (detailed in Supplementary Fig. 4)[54]. Briefly, a 637 nm multimode diode laser (RLM-6000L, Kvant lasers, Slovakia) was fiber-coupled before homogenization (Albedo system, Errol, France), generating a top-hat profile. Following collimation, the beam was focused at the back focal plane of the objective (100× Plan-Apo 1.45 NA, Nikon, Japan). A polarizing beam splitter transmits plane-polarized light with a specific orientation to the quarter-wave plate, which converts it to circularly polarized light. When used together, these components function as an optical isolator, efficiently separating the signal of interest from the illuminating light. Light reflected back from the sample-glass interface interferes with the light scattered by the objects within the sample, was then separated from the incident light, and directed onto a high-speed CMOS camera (PCO.dimax CS1, ExcelitasPCO GmbH, Germany). Focus control is then provided by a piezo-stage (P-545-3R8S, Physik Instrumente, Germany). The camera, laser, and piezo-stage are controlled using a custom LabVIEW program (National Instruments, USA).

Glass coverslips (24 × 60 mm, #1.5 thickness, Epredia) were cleaned by sequential sonication in chloroform, acetone, and isopropanol for 15 min before drying with $N_2$. Circular silicone spacers ($\phi$9 × 2.5 mm thickness, Merck) were washed with the same protocol, dried under vacuum, and then placed on top of the cleaned coverslip to form a chambered coverslip (Supplementary Fig. 4). The reaction chamber was sealed with an additional coverslip (22 × 22 mm, Merck) which was cleaned using the same procedure. To prepare plasma-cleaned coverslips, the solvent-cleaned coverslips were treated with oxygen plasma for 6 min (Diener Electronic, Femto, Germany).

$PCL_{45}$:$PCL_{45}$-b-$PDMA_{348}$ platelets synthesized in bulk were spin-coated onto the cleaned coverslip at 3200 rpm for 15 s until the desired surface density was achieved. Then a silicone spacer was placed onto the platelet-coated surface before the addition of methanol: water (30:70, v: v) solution and imaging. A laser power density of 2 μW μm$^{-2}$ at 637 nm and a camera exposure time of 900 μs were selected unless otherwise stated. A minimum number of 100 platelets were analyzed to determine the average size.

## Data analysis

Image analysis of raw images collected from iSCAT microscopy experiments proceeded via the following steps: (1) Dark counts subtraction was conducted by subtracting each frame by a frame recorded under the same conditions without laser illumination; (2) Temporal fluctuations in laser intensity were removed via the division of each frame by the frame modal pixel value; (3) Background correction was then performed via subtraction of a background corresponding to the median-average of 10 frames corresponding to the image area prior to platelet growth; (4) A binary mask was created from a low-pass Gaussian filtered ($\sigma = 2$ px) replica of the image and used to isolate individual platelets; (5) Platelet parameters, including area, long and short axis length, aspect ratio, and perimeter were then collected using the built-in Analyze Particles function in the Fiji image analysis software[73].

## Data availability

The raw data generated in this study have been deposited at kcl.figshare.com doi:10.18742/28309298. Data is also available from the authors on request.

## Code availability

The main data analysis was written in Python 3.6 and used standard Python packages. The code for each analysis can be provided together with the raw datasets from the corresponding author.

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

## Acknowledgements

Y.G. thanks the King's-China Scholarship Council PhD Scholarship program for funding. M.I.W. is funded by the Wellcome Trust (224327/Z/21/Z). This work was supported by the BBSRC (BB/R001790/1). T.X., J.Y.R., and R.O.R. thank the University of Birmingham for financial support.

## Author contributions

M.I.W. and R.O.R. conceived this work. Y.G. and T.X. contributed to this project equally, they performed the experimental studies and data analysis. V.W., Y.X., J.Y.R., and L.X. participated in the project discussion and commented on the manuscript.

## Competing interests

The authors declare no competing interests.
