## [Transparent Peer Review file · Nature Communications]

Real-time label-free imaging of living crystallization-driven self-assembly

Corresponding Author: Professor Mark Wallace

Version 0:

Reviewer comments:

Reviewer #1

(Remarks to the Author)

Guo et al reports the use of iSCAT technique to quantitative tracking of crystallisation of copolymers in real time. The results claims to provide insight into polymer composition in real time that is otherwise missed out by TEM or even fluorescence techniques. There are several major issues with the manuscript

1) Whilst the overall application is interesting and potentially useful, the imaging quality quantification of iSCAT requires significant improvement before it can be consider for publication. The degree of contrast and calibration metric for iSCAT is not fully reported. Judging from the results, the contrast is below existing iSCAT standards (SNR).

2) The paper only applies iSCAT to measure crystal growth, but do not provide clear use of how iSCAT's nanometric axial precision to measure 3D growth rates. The true use of iSCAT is axial profiling and measurement label-free and not lateral. The measured area (shown in Fig.4 for exampl) do not need iSCAT. In fact, phase contrast or DIC could also generate the same outcome. More importantly, even other forms of quantitative phase hologram can be use. It is unclear why iSCAT supersede other form of nanoscale label-free techniques.

Generally, the novelty factor of just application of a existing is not sufficient strong to be consider for publications. The discussion of iSCAT and calibration will be critical to prove the use of iSCAT over other label-free techniques.

Reviewer #2

(Remarks to the Author)

[Note from the Editor: Please also see attached PDF]

This manuscript describes the tracking of the growth of PCL-b-PDMA 2D platelets prepared by living crystallization-driven self-assembly (CDSA). The most common methods currently available for tracking particle growth are generally limited to their static nature, poor resolution and robustness. In this work, the authors demonstrated that iSCAT represents a tool that allows the tracking of particle growth at a high spatio-temporal resolution. It is also able discern the growth of multi-layered CDSA particles with different corona chemistry. The live imaging data collected are impressive and I believe this tool will provide further exploration into better understanding of living CDSA, that will in turn enable the preparation of CDSA particles with more complex and functional chemistry. However, the experiments conducted in this work are not numerous. Bigger sample sizes are needed, along with proof-of-concept experiments to show that the kinetic parameters obtained by using iSCAT are comparable with the literature findings, or if different, why.

Additionally, the authors attempt to assert the potential of iSCAT as a technique to better understand particle growth in CDSA. However, not much explanation is provided on the method itself, including the microscope setup, components needed, how it's different to general brightfield microscopy, data collection and processing etc. How was segmentation carried out? As I can see, the images looked like typical bright field images with enhanced contrast and are different from the

images collected from interferometry. How are they different? While I believe the work shown in this paper to be important for the field, I do believe that more work is required to solidify the findings and demonstrate the general applicability, reliability, and robustness of the method. Therefore, I would recommend reconsidering this manuscript after major revision.

1. Page 4 paragraph 1:

- The authors used ms to describe temporal resolutions for liquid TEM and AFM but used MHz for iSCAT. Please keep unit the same for consistency.
- What is the spatial resolution for iSCAT?
- The authors mentioned here that iSCAT would be good to study the growth of 1D cylindrical fibres prepared by CDSA. How is this translatable to tracking the growth of 2D platelets? And can this technique be used to track the self-assembly process of particles prepared by other types of solution phase self-assembly?

2. Page 5 – Real time monitoring of platelet formation:

- Please describe clearly how the seeds were prepared and provide the size characterization data. Include a brief explanation in the results and discussion section.

3. Page 5 paragraph 1:

- “custom built microscope” – please elaborate what this means. Provide more details and discuss.

4. Page 5 paragraph 2:

- Please describe unimer concentration with unimer:seed ratio following the general format of reporting seen in CDSA papers
- What happens when unimer concentrations used are above and below 0.32 μM ? How was this optimized?
- Why was the unimer concentration increased when a higher frame rate was used?

5. Page 6 Figure 2:

- The authors claimed that iSCAT is able to track the growth of each individual particle. However, the data reported in Figure 2B only represents 11 particles out of the presumably hundreds of thousands of particles in the bulk solution. How is the data collected by iSCAT representative and statistically significant with respect to the rest of the sample? At the moment the sample size is too small. Multiple repeats and ROIs are needed to improve the reliability of this technique in reporting the dynamic nature of seeded growth.

6. The supplementary movies showing the seeded growth of platelets are very impressive, especially the growth of different layers. Can the authors identify the presence of the seeds at the start? From my viewpoint, the seeds are not visible on iSCAT. How can the authors confirm that that 2D platelet formation seen recorded via iSCAT is indeed seeded growth and not an effect of self-nucleation? Control experiments without the presence of seeds are needed.

7. Page 6 last paragraph:

- What is the “ACM” in ACM-PCL and ACM-PCL-b-PDMA?

8. Figure 3B – it is quite hard to identify the mean and SD of the data. Please change the format to reveal the mean and SD by using for example box and whiskers plot showing the individual data points.

9. Platelet growth kinetics:

- Can the authors validate the kinetics findings with other means of characterization methods and report how the kinetics data obtained by iSCAT is different or more reliable and precise compared to existing techniques?
- Is there a reason why the authors chose to observe the growth of 2D platelets instead of 1D fibres? I think this technique is a great tool to better understand the mechanism of living CDSA and has the potential to open up many opportunities that will allow further expansion. However, I would like to first see how iSCAT is being used to probe the growth kinetics of 1D cylindrical particle, and how the results compared to previous studies (eg. Manners and co-workers', ACS Nano 2018; Choi and co-worker Chem. Sci. 2020).
- It would also be interesting to see the versatility of iSCAT in probing the growth of CDSA particles made from other types of core-forming blocks like PLA or polycarbonates.

10. It is unclear what solvent system was used in the self-assembly system investigated. How versatile and robust is iSCAT to different solvents (eg. changes in refractive indices and corona swelling in different solvent conditions)? Since it has been used to track actin polymerization, I assume it works in aqueous condition too?

11. Figure 5G – this is a similar concern as mentioned above. This figure shows the area evolution of 3 platelets. Can we derive the kinetics of particle growth based on such a small sample size? How can we be certain that we are representing the entire bulk population?

12. Conclusion

- The authors claimed that iSCAT can also be used as ex situ characterization of particle morphology due to its fast data collection, low cost, simple sample preparation and high throughput. However, up to this point, I don't feel like I learned much about iSCAT as a technique itself. I suggest the authors to provide explanation on the technique and the equipment involved in setting it up.
- Additionally, I had a brief experience working with interferometry, and I learned that it is not as simple to set up and it actually takes a lot of optimization to be able to get the scattering data.
- I don't think I agree with the claim “high throughput” here. As mentioned previously, the data showed here only represents a sample size of less than 30, which is in itself not considered to be statistically significant.

Reviewer #3

(Remarks to the Author)

The authors provide a novel methodology for observing living crystallization-driven self-assembly (CDSA) using iSCAT. This enables them to probe individual crystal formation in-situ – with high temporal/spatial resolution compared to other existing techniques. The utility of iSCAT for doing so is demonstrated relative to other methodologies (TEM, AFM, and CLSM). Further exploration is done with the application of a kinetically derived model to examine growth dependencies on

different technical parameters. Commentary on the significance of these parameters and additional multi-layer observational data completes the technical demonstration of iSCAT.

The work presents a very nice and interesting case study that can set the stage for a more widespread use of iSCAT in material science and molecular physics. However, the paper neither presents a new aspect of iSCAT as a method (hardware or analysis), nor does it report a substantially new insight into crystal growth as a whole or in the specific case of the material used. In particular, although the authors list a number of advantages that iSCAT has for studying assembly and growth phenomena, they do not present a study that exploits these in a decisive manner. It remains unclear if the results could not have been obtained from -AFM measurements. As a result, the reviewer cannot recommend publication in Nature Communications. The paper is deemed to be more appropriate for publication in one of the high-impact ACS journals. Below, we provide some more detailed comments that can be considered when revising the manuscript.

Major Comments:

- As the authors point out, the strength of iSCAT is its high temporal resolution and sensitivity. To this end, the reader expects to see an emphasis on these aspects also in the study at hand. For example, it would be particularly interesting to zoom into the early growth of the nanocrystals.
 - Another unique advantage of iSCAT is its sensitivity and resolution in the axial direction, which can be used to rival the AFM height measurements. It would be interesting to plot the change of iSCAT contrast as a function of time at a given point of the crystal, e.g. in Fig. 5B,F.
 - Related to the previous point, the authors use the area or the lateral dimensions as a metric for the growth process. It would be helpful if they could say something about the change of the crystal thickness. In this context, it would also be important to know how thick the original seed nanocrystals are.
 - What is the significance/meaning of the periodic ring-like variations of contrast in Fig. 5B? Could it be that the newer layers hang over the old ones? The current presentation suggests that the growth is always in contact with the substrate.
 - Page 4, the authors state "To date, iSCAT has been applied predominantly to the study of biological systems." I suggest to change this to "... the study of assembly in biological systems". iSCAT has indeed also been used in other contexts. The authors should cite a recent example that appeared in ACS Photonics 11, 737 (2024), which is also a good example for the power of iSCAT in studying small height variations.
 - Page 7, First Paragraph: A great deal of effort is made to note the deficiencies in other observation techniques culminating in the comparison of Figure 3. Would the authors please make clearer how the observational data in Figure 3 relates to the time resolution of the other techniques and more clearly delineate the unique advantages of iSCAT?
 - Page 10, First Paragraph: Author speculation for the anomalous data noted in Fig 4C is too brief. Please expand upon this supposition (evidence for 5% THF content being the point where significant solubilization occurs, or other evidence to support this claim) to better improve the real data outcomes from this study.
 - Page 10, Second Paragraph: The observed elongation of the platelets as a function of THF concentration deserves some exploration. What is the significance of this observed fact? Why with solvent concentration? Previous observation of axial growth heterogeneity in living CDSA? All of these should be considered here even if the answer is that it's unknown and warrants further study.
 - Page 13, Second Paragraph: The declaration of sub-millisecond temporal resolution does not seem properly supported or otherwise discussed within the main body of the work. This is particularly confusing when the stated acquisition frequency in the methods is single digit Hz. Additional context for this claim, and/or clarification of its origin should be stated/restated here
- Minor:
1. Page 5, Third Paragraph: Please clarify the use of the words "final platelet area". Final as in the asymptote converged to in figure 2B, or an arbitrary timestep?
 2. Page 8, First Paragraph: Please re-reference Gao (ref 53) when stating the assumption that parameter "B" can be held constant (as per their supplementary material).
 3. Figure 4: A legend for each column of information would aid in re-referencing the content of the graphs much more quickly.
 4. Figure 4: Please add language to indicate that the error bars are representing standard deviation of the area distribution in all plots (with error bars present, assuming that is true). Additionally, please state the number of platelets analyzed for each point (100) in the same statement as the standard deviation (earlier in the caption than its current form).
 5. Figure 4: Either here or in the supplemental please add some explanation for how the various time points have been chosen for these plots. If done in the supplemental please reference here.
 6. Figure 5F: Please clarify the boundaries that are "less distinct, albeit still identifiable" in some form in this figure. Suggestion: a secondary series of images with rough boundaries overlaid on each image (color coded as each layer is added)
 7. In the conclusion section, it is stated that iSCAT is limited to studies close to a surface. Although most iSCAT studies are performed in this mode for the convenience of higher mechanical stability, it is also possible to use an external reference for iSCAT. This is pointed out in Fig. 1d of Ref. 41 and used in some measurements of Ref. 43.

Reviewer #4

(Remarks to the Author)

Version 1:

Reviewer comments:

Reviewer #1

(Remarks to the Author)

Guo and co-workers provided a revised manuscript for the application of iSCAT to map 2D real time mapping of crystallization. The revised manuscript fails to take into the account and improve the 3D quantitative nature of iSCAT. Whilst I appreciate that the image quality of the revised manuscript has improved, this is a missed opportunity to address 3D quantitative refractive index (RI) measurement of iSCAT. Lateral resolutions of iSCAT is not below the diffraction limit, in fact, it is at the diffraction limit. However, iSCAT has higher degree of sensitivity to measure small optical path length difference (RI) in aqueous solutions (dielectric objects).

Sensitivity of iSCAT is an indication of the path length difference between the scattered light from the object and the reflected light from a reference (coverslip). Looking at Fig.5B and Fig. S16, the interference fringes (resembling Fresnel rings) are clearly visible from the iSCAT images. The authors claims that this is a multi-annulus platelet formation, but the interference fringes can also be a result of varying path-lengths (refractive index) within the platelet which is a volumetric measurement. I would strongly urge the author to carry out additional 3D quantitative measurement that can be used to determine the refractive index changes in the platelet formation. Without additional axial measurement, it will be challenging for readers to appreciate the quantitative refractive index imaging of nanocrystals growth using iSCAT.

Below are For Reviewer.2

1. Fig 2C and D only included additional data (40 platelets). Whilst this increase sample size, there is lack of discussion on statistic significant. At the moment, Fig. 2C should include variance of the area over time instead of scattered plots.
2. satisfied iSCAT explanation included.
3. Fig. S1 attempts to discuss the background subtraction. However, the background subtraction here do not appears to be iSCAT specific (<https://pubs.acs.org/doi/full/10.1021/acsp Photonics.7b00238>). In my view, the background subtraction in iSCAT requires a more indepth estimate of background noise. This is lacking in the supplementary results that need to be done so that it is consistent with modern iSCAT systems. It is important to bear in mind that iSCAT images are interferometric images (intensity fringes) which are not standard intensities only images.
4. I am able to able comment on this. Because I believe the authors have not developed more insights from iSCAT images which is axial information (thickness). As such, i think that the revised manuscript need to reflect that.
5. satisfied
6. satisfied
7. satisfied
8. It is still unclear the real time aspect. What is consider real time in platform formation from seed to imaging...A imaging protocol in supplementary can help.
9. satisfied
10. satisfied
11. satisfied
12. satisfied
13. satisfied
14. satisfied
15. satisfied
16. satisfied
17. satisfied
18. satisfied
19. satisfied
20. The authors has not provide clear discussion of Refractive index change in platelets which can affect iSCAT results.
21. satisfied
22. satisfied
23. The authors fail to discuss about vibration isolation which is what is needed to be discussed here, particular for interferometry.
24. satisfied

Reviewer #3

(Remarks to the Author)

The authors have performed substantial revisions to the manuscript and SI and have addressed all my major concerns. The manuscript can be published as is.

Reviewer #4

(Remarks to the Author)

Version 2:

Reviewer comments:

Reviewer #1

(Remarks to the Author)

1) Value of RI versus 3D imaging:

The revised manuscript clarifies that 3D information derived from iSCAT is not valuable in this study. However, the authors state, "The refractive indices of PCL and PDMA are also similar ($n_{PCL} = 1.476$; $n_{PDMA} = 1.430-1.456$)." This is an inaccurate statement. Interferometry, particularly with iSCAT, can detect refractive index (RI) differences as small as 0.001. Therefore, suggesting that the RI difference between PCL and PDMA is similar is an understatement. It is important to note that iSCAT measures optical path length differences because of either refractive index or thickness at the nanoscale. The real challenge lies in decoupling the effects of thickness and RI. I recommend rephrasing the statement to reflect this more accurately.

Furthermore, the authors should address the practical resolution of this issue, which can be evaluated through AFM methods (as shown in their supplementary). Including this information in the main text can be valuable. It will help provide a clearer explanation of the challenges involved in distinguishing between thickness and RI variations.

2) Clarification on Thickness vs. RI:

The manuscript states, "This could be due to a difference in refractive index or a difference in thickness," and later clarifies, "we attribute the change in contrast to be primarily a thickness effect." However, this explanation may be causing confusion. If the effect is related to thickness, 3D measurements would indeed be necessary to resolve the variation. On the other hand, changes in refractive index could be linked to the crystallization process.

To avoid this ambiguity, I urge the authors to provide a more detailed discussion of the relationship between thickness and refractive index in this context. It would be helpful to outline potential future work focused on decoupling the effects of thickness and refractive index. I also appreciate that although interferometric signals may be less quantitative, they provide valuable insight for routine nanoscale inspection.

Figure S19: Phase imaging - this should include a colormap indicate either phase in radian or pathlength difference

Reviewer 1

1. Whilst the overall application is interesting and potentially useful, the imaging quality quantification of iSCAT requires significant improvement before it can be considered for publication. The degree of contrast and calibration metric for iSCAT is not fully reported. Judging from the results, the contrast is below existing iSCAT standards (SNR).

We have updated our manuscript to provide a dedicated Supplementary Methods section describing our protocol for iSCAT calibration and quantification of contrast in detail. Our methods for contrast calibration have also recently been published in detail; we now cited that work directly.

PS8.L862: "Instrument calibration and contrast extraction..."

We have also discussed how recent improvements in iSCAT instrumentation might be applied to our work.

P18.L313: "In addition, there remains considerable scope to further enhance our detection sensitivity; for example, by applying recent reports of machine learning⁶⁷ and optimization of reflected vs. scattered signals⁶⁸ to improve iSCAT mass sensitivity."

2. The paper only applies iSCAT to measure crystal growth, but do not provide clear use of how iSCAT's nanometric axial precision to measure 3D growth rates. The true use of iSCAT is axial profiling and measurement label-free and not lateral. The measured area (shown in Fig.4 for example) do not need iSCAT. In fact, phase contrast or DIC could also generate the same outcome. More importantly, even other forms of quantitative phase hologram can be use. It is unclear why iSCAT supersede other form of nanoscale label-free techniques.

We appreciate iSCAT can be used for 3D tracking, however here our focus is not on 3-dimensional particle growth, but rather the 2D control of particle morphology and kinetics afforded by CDSA. We agree that iSCAT can provide information regarding self-assembly without the need for 2D area measurement. We have expanded our dataset to include this point and have provided expanded contrast vs. time data for early growth stage of CDSA (Figs. 2E & F) to highlight this fact.

P9.L148: "During the early stages of CDSA, the platelet size is below the diffraction limit, and thus platelet area cannot be used to examine growth kinetics.. ..."

Our focus on 2D morphology and area growth is driven by the challenge of better understanding the kinetics and design rules for CDSA assembly. We acknowledge that Phase Contrast or DIC techniques could also be utilized to gather this information and have cited previous works employing these methods to study 2D structures in our introduction. However, it is the enhanced sensitivity of iSCAT compared to these methods that provides deeper insights. In our updated manuscript we have emphasized our use of iSCAT to observe the early stages of CDSA, when the platelet size is below the detection limit of PC and DIC.

P4.L50: "Label-free techniques such as phase contrast or differential interference contrast microscopy can circumvent this issue but are typically limited in detection sensitivity.³⁸⁻⁴¹ ..."

P9.L148: "During the early stages of CDSA, the platelet size is below the diffraction limit, and thus platelet area cannot be used to examine growth kinetics.. ..."

We have also updated our discussion to emphasize iSCAT's high sensitivity in axial profiling and its future potential to track 3-dimensional self-assembly.

P18.L309: “Here we have focused on 2D CDSA, but the interferometric nature of iSCAT also embeds axial information; enabling, for example, the measurement of nanoparticle axial position and size.^{47,48,61–63} ...”

3. Generally, the novelty factor of just application of an existing is not sufficient strong to be consider for publications. The discussion of iSCAT and calibration will be critical to prove the use of iSCAT over other label-free techniques.

We would emphasize that it is the not our use of iSCAT per se, but rather its application to living crystallization-driven self-assembly that we believe is of importance. Conventional techniques such as TEM, AFM, and confocal microscopy have significant limitations in capturing the dynamic evolution of self-assembled structures in real-time. In our work, we address this challenge to monitor living crystallization-driven self-assembly process in real-time. To address the overall criticism, we have revised our introduction and conclusion to aid the reader in understanding the context of both the microscopy method and the application to polymer nanostructure control. Together, we trust this provides improved context to assess the significance of applying iSCAT microscopy to CDSA.

P4.L63: “iSCAT relies on the interference between light scattered from an object of interest supported on a glass coverslip, and reference light reflected back from the same glass-sample interface (Fig. 1B)...”

P18.L317: “Overall, iSCAT is a potent technique that offers a new perspective on improving CDSA-based nanoparticle engineering. Here, we have demonstrated the applicability of iSCAT to study PCL-based 1D fibers and 2D platelets....”

Reviewer 2

1. the experiments conducted in this work are not numerous. Bigger sample sizes are needed, along with proof-of-concept experiments to show that the kinetic parameters obtained by using iSCAT are comparable with the literature findings, or if different, why.

We fully agree with this comment. To address this, we have significantly expanded our datasets and updated the results in Figs. 2C & D. Additionally, we have included discussions comparing our kinetic results with previous studies in the revised manuscript.

P8.L138: “Analysis of the time evolution of platelet area yields the kinetics of individual platelet growth (Fig. 2D). 210 platelets were recorded from 3 experimental repeats over a 25 min time-lapse. The individual growth trajectories of 40 representative platelets are shown in Fig. 2D....”

P6.L114: “Comparing these data to previous work on poly(ferrocenyldimethylsilane)-b-polydimethylsiloxane (PFS₆₃-b-PDMS₅₁₃) 1D fibers (10 to 30 mg/mL unimer and 0.1 mg/mL seed), we observe faster kinetics in our PCL-based system ($5.1 \times 10^{-3} \text{ s}^{-1}$ compared to 1.8×10^{-4} to $2.13 \times 10^{-4} \text{ s}^{-1}$), despite lower unimer and seed concentration.”

P14.L231: “Compared with 1D fiber formation using a poly(ferrocenyldimethylsilane)-based core-forming blocks³⁰ or 2D nanosheet formation from poly(cyclopentenylene vinylene)-based core-forming blocks,³² our PCL-based system exhibits faster kinetics in both 1D and 2D assembly ...”

2. Additionally, the authors attempt to assert the potential of iSCAT as a technique to better understand particle growth in CDSA. However, not much explanation is provided on the method itself, including the microscope setup, components needed, how it's different to general brightfield microscopy, data collection and processing etc.

We agree. A detailed description on iSCAT microscopy instrumentation has now been included in the Experimental Methods section. This includes information on the microscope setup and required components.

P24.L469: "A wide-field iSCAT was previously constructed (detailed in Fig. S3).⁵⁴..."

In addition, data collection and processing methods, and method for image segmentation was also included in Supplementary Method section.

PS6.L843: "Data analysis---in-situ iSCAT monitoring of CDSA process..."

We have also amended our introduction to include a better introduction to iSCAT theory and its distinctions from general brightfield microscopy. And included contrast determination and instrument calibration in the Supplementary Method section.

P4.L63: "iSCAT relies on the interference between light scattered from an object of interest supported on a glass coverslip, and reference light reflected back from the same glass-sample interface (Fig. 1B)..."

P4.L67: "For a diffraction-limited object, iSCAT signals appear as an Airy disc of concentric dark and light rings caused by the interference between the reflected and scattered signals...."

PS8.L862: "Instrument calibration and contrast extraction...."

3. How was segmentation carried out? As I can see, the images looked like typical bright field images with enhanced contrast and are different from the images collected from interferometry. How are they different?

We have expanded our description of our data segmentation method as a new Supplementary Method section. Our expanded introduction to iSCAT mechanism, alongside more careful figure captioning, we trust will address the queries regarding interpretation of these images.

PS6.L843: "Data analysis...in-situ iSCAT monitoring of CDSA process..."

P4.L63: "iSCAT relies on the interference between light scattered from an object of interest supported on a glass coverslip, and reference light reflected back from the same glass-sample interface (Fig. 1B)..."

P4.L67: "For a diffraction-limited object, iSCAT signals appear as an Airy disc of concentric dark and light rings caused by the interference between the reflected and scattered signals...."

4. While I believe the work shown in this paper to be important for the field, I do believe that more work is required to solidify the findings and demonstrate the general applicability, reliability, and robustness of the method. Therefore, I would recommend reconsidering this manuscript after major revision.

We agree that a strengthened demonstration of the general applicability of our method would improve the manuscript. We have worked to significantly expanded the number of measurements supporting in our 2D CDSA datasets. We now also include additional work demonstrating the method for a different

system – measuring 1D CDSA fiber growth kinetics using our method and validating these results with parallel AFM measurements.

P8.L140: “210 platelets were recorded from 3 experimental repeats over a 25 min time-lapse. The individual growth trajectories of 40 representative platelets are shown in Fig. 2D.....”

P6.L102: “1D fibers.....”

P11.L188: “We first validated our method by ensuring that the time-dependence of platelet morphology sampled from a CDSA reaction at predetermined intervals and then analyzed by both iSCAT and AFM was consistent (Fig. S12).....”

5. Page 4 paragraph 1: The authors used ms to describe temporal resolutions for liquid TEM and AFM but used MHz for iSCAT. Please keep unit the same for consistency.

We agree with this comment, consistency is important for clarity. We have harmonized all measurement units through the manuscript.

e.g. P9.L159: “Data were collected with a temporal resolution of 333 μ s (Fig. S10, Movie S4).”

6. What is the spatial resolution for iSCAT?

This information now has been added to the introduction of the revised manuscript.

P4.L69: “iSCAT offers lateral and axial resolutions of around 200 - 300 nm and 10 - 100 nm, respectively...”

7. The authors mentioned here that iSCAT would be good to study the growth of 1D cylindrical fibers prepared by CDSA. How is this translatable to tracking the growth of 2D platelets? And can this technique be used to track the self-assembly process of particles prepared by other types of solution phase self-assembly?

We have included these data, using iSCAT to study 1D cylindrical fibers and expanded this explanation on the straightforward application of iSCAT to other self-assembly processes. We have also revised our introduction to expand the list of examples where iSCAT was previously used to monitor assembly of other biological systems.

P6.L102: “1D fibers.....”

P8.L120: “Following our initial imaging of 1D fiber formation, we next focused on 2D platelet growth, where the characteristic morphology and fast controllable assembly of platelets might be used to assess the capabilities of iSCAT for CDSA measurement.....”

P4.L56: “To date, iSCAT has mainly been used to study biological systems, including the mass measurement of individual proteins, the formation of lipid membranes, and biological diffusion using single-particle tracking of metal nanoparticles.⁴⁵⁻⁴⁷....”

8. Page 5 – Real time monitoring of platelet formation: Please describe clearly how the seeds were prepared and provide the size characterization data. Include a brief explanation in the results and discussion section.

A more detailed seed preparation protocol is now included in the Experimental Method section.

P21.L387: “Preparation of PCL₄₅-b-PDMA₃₄₈ seed solution...”

Size characterization data has been added in the supporting information.

PS10.L883: “Figure S2. Seed characterization.”

In the discussion section, a brief explanation is now included.

P5.L93: “Uniform seeds were prepared from polydisperse fibers of PCL₄₅-b-PDMA₃₄₈ formed in ethanol (Fig. S2A), followed by sonication to produce short fibers with consistent size and length (mean length 24.6 ± 4.9 nm, Figs. S2B & C)....”

9. Page 5 paragraph 1: “custom built microscope” – please elaborate what this means. Provide more details and discuss.

We have expanded our explanation in the description and provided expanded Experimental Method section describing our instrument.

P24.L469: “A wide-field iSCAT was previously constructed (detailed in Fig. S3).⁵⁴...”

10. Page 5 paragraph 2: Please describe unimer concentration with unimer:seed ratio following the general format of reporting seen in CDSA papers.

Please note that now Fig. 2 has also been revised to provide larger data sets (and include the 1D cylindrical fiber formation) as requested by the reviewer 2.

To conduct this experiment, we immersed our plasma-cleaned coverslip into a poly(ϵ -caprolactone)-*b*-poly(2-vinylpyridine) (PCL-*b*-P2VP) seed ethanol solution, in order to attach P2VP seeds onto the surface (see more detail in Experimental Methods section). As a result, unimer:seed ratio cannot be given in terms of a ratio of two solution concentrations, but rather it depends directly on the 2-dimensional surface density of seeds, and hence the area of sample covered. We have updated the manuscript to report both unimer concentration and seed surface density.

Reviewer 2’s request for a general format of reporting seen in CDSA papers might also be interpreted as requesting mass concentrations (i.e., mg/mL) rather than the molarities we report. We have additionally changed our units to mass concentrations, but for clarity and accessibility across fields, we have elected to also report the molecular measure provided by molarities.

P8.L122: “Seeds were deposited on glass coverslips at a surface density of $\sim 0.16 \mu\text{m}^{-2}$”

11. What happens when unimer concentrations used are above and below 0.32 μM ? How was this optimized? Why was the unimer concentration increased when a higher frame rate was used?

As shown in our kinetic study (Fig. 4A), higher unimer concentrations lead to faster kinetics. The unimer concentration was selected based on the rationale behind a specific experiment. For example, for in-situ recording (Fig. 5G), a higher frame rate is desirable to capture more kinetic information for reaction conducted with higher a unimers concentration. We have included this information into our discussion.

P12.L198 “As expected, a higher initial unimer concentration leads to faster platelet growth, producing platelets with larger final surface area...”

P8.L126 “We selected a PCL₄₅:PCL₄₅-b-PDMA₃₄₈ unimer mixture concentration of 0.35 μM (3.33 $\mu\text{g}/\text{mL}$) in methanol to ensure slow growth of well isolated platelets. Keeping the seed surface density fixed, increasing the unimer concentration results in faster kinetics...”

12. Page 6 Figure 2: The authors claimed that iSCAT is able to track the growth of each individual particle. However, the data reported in Figure 2B only represents 11 particles out of the presumably hundreds of thousands of particles in the bulk solution. How is the data collected by iSCAT representative and statistically significant with respect to the rest of the sample? At the moment the sample size is too small. Multiple repeats and ROIs are needed to improve the reliability of this technique in reporting the dynamic nature of seeded growth.

We agree - it is important to emphasize more clearly the parallel nature of these measurements. We now expand the data sets presented in Figs. 2C & D. In these experiments, the same reaction conditions were measured over 3 repeats, with approximately 70 particles per experiment. Platelet area growth from 40 representative particles from that dataset is now reported in Figure 2D.

P8.L138: “Analysis of the time evolution of platelet area yields the kinetics of individual platelet growth (Fig. 2D). 210 platelets were recorded from 3 experimental repeats over a 25 min time-lapse. The individual growth trajectories of 40 representative platelets are shown in Fig. 2D...”

We understand the reviewer’s underpinning question here to be if in-situ data collected by iSCAT is representative for the rest of population. To address this, we compared platelet area evolution from data collected from in-situ recording and data collected from the same sample but varied position on the surface. The results shown in Fig. S6 support our conclusion that the data collected by in-situ iSCAT is representative of the rest of the sample.

P8.L141: “All platelets show similar growth kinetics and consistent final platelet area ($\sim 1.12 \mu\text{m}^2$) at the end of the recording (Fig. S9A). The area distributions of platelets collected at the end of in-situ recording (Fig. S9A) were independent of the position of the platelet in the sample (Fig. S9B). These data confirm the ability of living CDSA to produce uniform assemblies with controlled size and demonstrate that the in-situ iSCAT imaging captures growth kinetics representative of the entire population.”

PS17.L890: “Figure S9: Comparison of platelet size distributions: End of in situ recording vs. subsequent sampling...”

We have also compared CDSA assembly prepared in bulk, collected at predefined time points, and analyzed by both iSCAT and AFM imaging, here particle sizes and growth kinetics are in agreement. We include this data as Fig. S12.

P11.L188: “We first validated our method by ensuring that the time-dependence of platelet morphology sampled from a CDSA reaction at predetermined intervals and then analyzed by both iSCAT and AFM was consistent (Fig. S12).”

PS20.L893: “Figure S12: Comparative analysis of AFM and iSCAT in morphological and kinetic studies...”

Additionally, we would like to emphasise that the data presented in Fig 4 is derived not from in-situ high-speed recording of the growth of individual platelets immobilized on the surface, but rather from a time-series of aliquots taken from the bulk solution. Thus, we expect these kinetics to be representative of the bulk reaction.

P11.L193: “Platelets were prepared in bulk under various reaction conditions, with sample aliquots removed at predefined time points to fully capture the reaction kinetics.....”

14. The supplementary movies showing the seeded growth of platelets are very impressive, especially the growth of different layers. Can the authors identify the presence of the seeds at the start? From my viewpoint, the seeds are not visible on iSCAT. How can the authors confirm that that 2D platelet formation seen recorded via iSCAT is indeed seeded growth and not an effect of self-nucleation? Control experiments without the presence of seeds are needed.

Control experiments were conducted without the presence of seeds on the surface. We now include these results in the revised manuscript (Fig. S5). We also now include additional data and analysis for early stage growth (Fig. 2).

P8.L130: “Control experiments without the presence of seeds show no characteristic fast growth (Fig. S5A)....”

PS13.L886: “Figure S5: Control experiment: Reaction conducted in the absence of surface-coated seeds....”

P6.L109: “the size of PCL₄₅-b-PDMA₃₄₈ seeds is below the detection limit of our setup, thus they are not observable....”

15. Page 6 last paragraph: What is the “ACM” in ACM-PCL and ACM-PCL-b-PDMA?

We have expanded our explanation of this section and properly cite the reference that introduces the ACM preparation method.

P9.L167: “Assembly of fluorescently-labeled platelets was initiated by adding 10 μ L of 1 mM(10 mg/mL) aminochloromaleimide (ACM)⁵⁵ -coupled homopolymer and block copolymer unimer mixtures (ACM-PCL₄₅:ACM-PCL₄₅-b-PDMA₃₄₈, 1:1, w:w) into 1 mL 167 0.25 μ M(10 μ g/mL) PCL₄₅-b-PDMA₃₄₈ seed solution in ethanol, followed by mixing via shaking....”

16. Figure 3B – it is quite hard to identify the mean and SD of the data. Please change the format to reveal the mean and SD by using for example box and whiskers plot showing the individual data points.

We fully appreciate it is important to compare means and SD for this data. However, as our platelet areas are small and cannot be negative, it is perhaps not prudent to assume a priori normally distributed data - where a box and whiskers plot would indeed be most appropriate. Our preference is to provide the reader with the raw datapoints to give an unbiased graphical representation of our measurements. We have thus elected to keep Fig 3B unchanged in the main manuscript; but have included an additional figure (SI Fig. S11) with boxplots for comparison to address the reviewer’s point.

PS19.L892: “Figure S11: Comparison of 2D platelet characterization methods....”

17. Platelet growth kinetics: Can the authors validate the kinetics findings with other means of characterization methods and report how the kinetics data obtained by iSCAT is different or more reliable and precise compared to existing techniques?

We agree with these comments. To make a direct comparison between growth kinetics using different methods we have compared CDSA assembly by sampling aliquots from a reaction prepared in bulk, followed by analysis by both iSCAT and AFM imaging. We include this data as Fig. S12. The morphology

of the platelets reported by iSCAT is comparable with the results from AFM imaging. In addition, the kinetics of assembly extracted by iSCAT and AFM are indistinguishable ($k' = 0.0474$ and 0.0473 s^{-1} , respectively).

P11.L188: “We first validated our method by ensuring that the time-dependence of platelet morphology sampled from a CDSA reaction at predetermined intervals and then analyzed by both iSCAT and AFM was consistent (Fig. S12).”

PS20.L893: “Figure S12: Comparative analysis of AFM and iSCAT in morphological and kinetic studies...”

18. Is there a reason why the authors chose to observe the growth of 2D platelets instead of 1D fibers? I think this technique is a great tool to better understand the mechanism of living CDSA and has the potential to open up many opportunities that will allow further expansion. However, I would like to first see how iSCAT is being used to probe the growth kinetics of 1D cylindrical particle, and how the results compared to previous studies (eg. Manners and co-workers', ACS Nano 2018; Choi and co-worker Chem. Sci. 2020).

The reviewer makes an excellent point regarding 1D vs. 2D CDSA growth. Indeed, our preliminary experiments originally began with examining 1D growth, before we chose to switch to 2D platelets for their more rapid kinetics of self-assembly, clear morphology, and ease of modification. We now include this preliminary data (modified Figs. 2A & B) to enable a better discussion of the differences in kinetics and emphasize the broad applicability of our method to study CDSA. We have made significant modifications to the manuscript text to incorporate these changes.

P6.L107: “Fig. 2A depicts a representative time series of iSCAT images collected during the formation of 1D fibers (637 nm, $4 \mu\text{W} \mu\text{m}^{-2}$, 400 μs exposure time, 3 s^{-1} per frame time-lapse). Upon unimer addition, uniform fiber elongation can be monitored as the reaction proceeds (Movie S1).....”

P8.L120: “Following our initial imaging of 1D fiber formation, we next focused on 2D platelet growth, where the characteristic morphology and fast controllable assembly of platelets might be used to assess the capabilities of iSCAT for CDSA measurement.....”

As requested, we also compare our growth rates for 1D CDSA with those reported by previous studies. Manners and co-workers reported a rate constant for cylindrical growth of 1.8×10^{-4} to $2.13 \times 10^{-4} \text{ s}^{-1}$ under conditions of unimer and seed concentration: 10 to 30 mg/mL and 0.1 mg/mL, respectively. Applying the kinetic model reported by Manners, our PCL-based system displays a somewhat faster kinetics with a rate constant of $5.1 \times 10^{-3} \text{ s}^{-1}$ under lower unimer and seed concentration. We have modified the manuscript to include a discussion of this comparison and updated our supplementary methods to describe our methods and analysis of this data.

P6.L114: “Comparing these data to previous work on poly(ferrocenyldimethylsilane)-b-polydimethylsiloxane (PFS₆₃-b- PDMS₅₁₃) 1D fibers (unimer and seed concentration: 10 to 30 mg/mL and 0.1 mg/mL, respectively), we observe faster kinetics in our PCL-based system ($5.1 \times 10^{-3} \text{ s}^{-1}$ compared to 1.8×10^{-4} to $2.13 \times 10^{-4} \text{ s}^{-1}$) despite lower unimer and seed concentration.....”

PS4.L804: “Kinetic model for 1D cylindrical fiber growth....”

PS6.L827: “Kinetic data fitting for 1D cylindrical fiber formation....”

19. It would also be interesting to see the versatility of iSCAT in probing the growth of CDSA particles made from other types of core-forming blocks like PLA or polycarbonates.

We now include results exploring the effects of replacing our core forming block with poly(η -octalactone) (POL). However at least for POL, the resulting particles do not exhibit the uniform morphology provided by PCL cores. This simple distinctive shape has been very helpful in the demonstration of the validity of our method. We therefore elected not to pursue additional measurements examining the effect of core variation. We report these observations in our revised manuscript (Fig. S6). We have expanded our discussion and conclusion to emphasize how this future area might be addressed.

P8.L131: “We also explored the effect of changing the core chemistry: Platelets prepared with a poly(η -octalactone) (POL)-based core-forming block exhibited less uniform morphology and size compared to the PCL-based system (Fig. S6)....”

PS14.L887: “Figure S6: In-situ iSCAT monitoring of poly(η -octalactone)-based CDSA platelets formation....”

20. It is unclear what solvent system was used in the self-assembly system investigated. How versatile and robust is iSCAT to different solvents (eg. changes in refractive indices and corona swelling in different solvent conditions)? Since it has been used to track actin polymerization, I assume it works in aqueous condition too?

Yes, the reviewer is correct that this method would work under aqueous conditions. To demonstrate our method is applicable to a wide variety of solvents and solution conditions we include Figs. S7 & S8; demonstrating CDSA iSCAT for platelet growth in ethanol, isopropyl alcohol, and methanol. The impact of solvent on iSCAT imaging was also explored in this figure by pre-assembling platelets before changing solvent.

P8.L134: “Furthermore, we observed changes in platelet morphology with solvent (Fig. S7), however, iSCAT was able to resolve CDSA assemblies under a wide range of solution conditions (Fig. S8).....”

PS15.L888: “Figure S7: Impact of solvent on platelet formation.....”

PS16.L889: “Figure S8: Impact of imaging medium on iSCAT imaging.....”

21. Figure 5G – this is a similar concern as mentioned above. This figure shows the area evolution of 3 platelets. Can we derive the kinetics of particle growth based on such a small sample size? How can we be certain that we are representing the entire bulk population?

Q21 has been answered in response to Q12.

22. The authors claimed that iSCAT can also be used as ex situ characterization of particle morphology due to its fast data collection, low cost, simple sample preparation and high throughput. However, up to this point, I don’t feel like I learned much about iSCAT as a technique itself. I suggest the authors to provide explanation on the technique and the equipment involved in setting it up.

We have updated our manuscript to include a comprehensive description of our method and equipment and the requirements for reproducing these measurements.

P24.L469: “A wide-field iSCAT was previously constructed (detailed in Fig. S3).⁵⁴...”

P4.L63: “iSCAT relies on the interference between light scattered from an object of interest supported on a glass coverslip, and reference light reflected back from the same glass-sample interface (Fig. 1B)....”

P4.L67: “For a diffraction-limited object, iSCAT signals appear as an Airy disc of concentric dark and light rings caused by the interference between the reflected and scattered signals.....”

23. Additionally, I had a brief experience working with interferometry, and I learned that it is not as simple to set up and it actually takes a lot of optimization to be able to get the scattering data.

Interferometry is typically very challenging due to the long path length between reference and sample arms of the interferometer. ‘Hands on’, that typically results in very difficult optimization and alignment issues that must be overcome. In contrast, for iSCAT the path length difference is determined by the separation of the sample and the reflected reference beam from the coverslip. This ‘common path’ approach with sub-micron differences in path length results in a very simple and stable approach to interferometry and is indeed why it can be applied to the complex samples and solution conditions important for CDSA. We update our manuscript to emphasize that point.

P4.L65: “Compared to conventional interferometry, the common path agreement of iSCAT provides a simple and robust approach to measure changes in light scattering from small objects.....”

We also now update our citations to include previous studies focusing on iSCAT construction from commercially available optical components, compatible with fluorescence microscopy.

P17.L300: “Additionally, this technique can be implemented using simple commercially available optical components compatible with fluorescence microscopy^{48,60}...”

24. I don’t think I agree with the claim “high throughput” here. As mentioned previously, the data showed here only represents a sample size of less than 30, which is in itself not considered to be statistically significant.

We have updated Fig. 2 to improve our data, and now plot kinetics from 40 representative particles (Fig. 2D). Overall, our time-lapse data (Fig. 2C) is collected from ~200 particles per 25 minutes without any efforts to properly parallelize the method or increase our field of view. At the very least we would argue that that is certainly not low throughput! However, the threshold required to claim ‘high’ throughput is open to interpretation; thus, we have elected to modify our manuscript to place any discussion regarding the high-throughput potential of this method into future work, rather than argue that this statement is directly supported by our existing data.

P8.L140: “210 platelets were recorded from 3 experimental repeats over a 25 min time-lapse. The individual growth trajectories of 40 representative platelets are shown in Fig. 2D.....”

P17.L298: “Beyond its capabilities for in-situ monitoring, iSCAT also holds much promise as a tool for ex-situ characterization of particle morphology due to its fast data collection, low cost, simple sample preparation, and high throughput.....”

Reviewer 3

1. As the authors point out, the strength of iSCAT is its high temporal resolution and sensitivity. To this end, the reader expects to see an emphasis on these aspects also in the study at hand. For example, it would be particularly interesting to zoom into the early growth of the nanocrystals.

We agree. To achieve this emphasis, we now report early-stage (i.e. diffraction limited platelet size) contrast evolution for individual platelets (Figs. 2E & F) and high-speed imaging (0.33 ms) of platelet growth (Figs. 2G & H). We have amended our discussion to emphasize these advantages.

P9.L148: “During the early stages of CDSA, the platelet size is below the diffraction limit, and thus platelet area cannot be used to examine growth kinetics. However, the evolution of particle contrast can be used to report on this early stage (Figs. 2E & F, Movie S3).....”

P9.L156: “iSCAT imaging can also be used to monitor platelet growth at higher time resolution (Figs. 2G & H).....”

P17.L280: “With its excellent sensitivity and sub-millisecond temporal resolution, iSCAT successfully provides detailed insights into the CDSA process, including early-stage growth and high-resolution imaging of the growth dynamics.....”

2. Another unique advantage of iSCAT is its sensitivity and resolution in the axial direction, which can be used to rival the AFM height measurements. It would be interesting to plot the change of iSCAT contrast as a function of time at a given point of the crystal, e.g. in Fig. 5B,F.

Axial resolution is an advantage of iSCAT that we have not significantly exploited in this study. However, our focus is on CDSA for 2 (and 1) dimensional layered materials, where we do not expect significant 3D information. The suggested plot by Reviewer 3, examining the time dependent change of contrast at a specific point for the data in Figs. 5B & F did not reveal any clear change. We can however see the change in contrast at the edge of the platelet (where there is clear height changes), and this is now reported in our revised (Figs. 2G & H).

P9.L156: “iSCAT imaging can also be used to monitor platelet growth at higher time resolution (Figs. 2G & H).....”

Given this comment, we also suspect there is some confusion over our use of the term ‘layer’ to refer to experiments with sequential addition of different unimer compositions: Different layers correspond to sequential addition of different unimer solutions as they are deposited at the growing platelet edge. However ‘layers’ might be wrongly imply epitaxial layers - building up on the flat 2D surface of a platelet. As our growth is edge-mediated we do not expect any gross axial (3D) changes in platelet morphology. To address this, we have replaced the term ‘layer’ in our revised manuscript with ‘annulus’, which more precisely describes a ring-shaped layer in a 2D plane. To mitigate any potential for confusion we have expanded our description on 2D edge-directed growth in the revised manuscript.

P14.L239: “Following CDSA, the platelet perimeter remains active for the templated edge-directed growth of additional platelet area.²⁶ Therefore, multi-annulus platelets can be prepared by sequential unimer addition.....”

The potential for future iSCAT measurements of CDSA to reveal axial information is however important as highlighted by the reviewer. We have therefore also expanded our discussion to include specifically the capabilities of iSCAT to examine 3D morphology for this application.

P18.L309: “Here we have focused on 2D CDSA, but the interferometric nature of iSCAT also embeds axial information; enabling, for example, the measurement of nanoparticle axial position and size.^{47,48,61–63} Consequently, we envisage iSCAT might also be applied to profiling more complex CDSA processes, with three-dimensional shape at high spatio-temporal resolution.^{64–66} ...”

3. Related to the previous point, the authors use the area or the lateral dimensions as a metric for the growth process. It would be helpful if they could say something about the change of the crystal

thickness. In this context, it would also be important to know how thick the original seed nanocrystals are.

The growth of a CDSA platelet is essentially 2 dimensional (e.g. Fig. 2C). For Figs. 5B-E, the unimer composition is varied with each annulus deposition. This difference in unimer mixtures varies both the composition ($PCL_{45}:PCL_{45}\text{-}b\text{-}PDMA_{348}$ vs. PCL_{45}) and the relative thickness of each annulus (Figs. 5D & E).

iSCAT contrast is dependent both on sample composition and thickness. We chose annuli of $PCL_{45}:PCL_{45}\text{-}b\text{-}PDMA_{348}$ and PCL_{45} to provide a clear change in sample thickness, as this is confirmed by our AFM measurements (Fig. 5E), thus we interpret our changes in iSCAT signal as being predominantly due to changes in annulus thickness.

P16.L250: "As iSCAT is sensitive to the change of path length difference between the scattered and the reference signals, this difference in contrast is likely due to the path length variation induced by the increased thickness of platelet annuli formed from $PCL_{45}:PCL_{45}\text{-}b\text{-}PDMA_{348}$ as compared to pure PCL_{45} . This interpretation is supported by the AFM measurement of two-annulus platelets prepared following the same protocol...."

Due to their small size (approximately 25 nm), the seeds themselves are below our detection limit; it is therefore not sensible to comment on seed thickness based on our iSCAT signal. We have clarified this in our revised manuscript.

P6.L108: "Upon unimer addition, uniform fiber elongation can be monitored as the reaction proceeds (the size of $PCL_{45}\text{-}b\text{-}PDMA_{348}$ seeds is below the detection limit of our setup, thus they are not observable. Movie S1)...."

4. What is the significance/meaning of the periodic ring-like variations of contrast in Fig. 5B? Could it be that the newer layers hang over the old ones? The current presentation suggests that the growth is always in contact with the substrate.

We believe this is confusion over our use of 'layers' to refer to edge-directed annual growth, whereas it might well be interpreted as edge-directed layering. Fig. 5B illustrates growth from the edge of platelets, wherein each successive layer envelops the preceding one.

The formation of periodic variation of iSCAT contrast is because we alternate the unimer composition used for each 'layer' of the CDSA platelet. As iSCAT is sensitive to the path length difference between the scattered signal from the sample and the reference light. Variation in sample thickness or composition would lead to the change of scattering signal (thus, the path length) which can be detected by iSCAT. The 'layers' with negative (dark) contrast were constituted of PCL/PCL-b-PDMA which is the mixture of homopolymer and copolymer, producing higher thickness after assembly. Meanwhile, 'layers' with lighter (less negative) contrast was formed from PCL, which is pure homopolymer, producing lower thickness after assembly.

We have replaced the term 'layer' with 'annulus' to avoid this potential confusion.

P14.L239: "Following CDSA, the platelet perimeter remains active for the templated edge-directed growth of additional platelet area.²⁶ Therefore, multi-annulus platelets can be prepared by sequential unimer addition....."

P16.L250: "As iSCAT is sensitive to the change of path length difference between the scattered and the reference signals, this difference in contrast is likely due to the path length variation induced by the increased thickness of platelet annuli formed from $PCL_{45}:PCL_{45}\text{-}b\text{-}PDMA_{348}$ as compared to pure PCL_{45}"

5. Page 4, the authors state “To date, iSCAT has been applied predominantly to the study of biological systems.” I suggest to change this to “... the study of assembly in biological systems”. iSCAT has indeed also been used in other contexts. The authors should cite a recent example that appeared in ACS Photonics 11, 737 (2024), which is also a good example for the power of iSCAT in studying small height variations.

Combining the suggestions from both reviewers 2 and 3, we have now revised this section and cited the recommended reference.

P4.L56: “To date, iSCAT has mainly been used to study biological systems, including the mass measurement of individual proteins, the formation of lipid membranes, and biological diffusion using single-particle tracking of metal nanoparticles.⁴⁵⁻⁴⁷....”

P18.L309: “Here we have focused on 2D CDSA, but the interferometric nature of iSCAT also embeds axial information; enabling, for example, the measurement of nanoparticle axial position and size.^{47,48,61-63} Consequently, we envisage iSCAT might also be applied to profiling more complex CDSA processes, with three-dimensional shape at high spatio-temporal resolution.⁶⁴⁻⁶⁶ ...”

6. Page 7, First Paragraph: A great deal of effort is made to note the deficiencies in other observation techniques culminating in the comparison of Figure 3. Would the authors please make clearer how the observational data in Figure 3 relates to the time resolution of the other techniques and more clearly delineate the unique advantages of iSCAT?

The time resolution presented in Fig. 3 is constrained not by the capabilities of iSCAT, but in the intent to robustly and directly compare different measurement techniques. In essence, for the data presented in Fig. 3, it shows that growth is completed well before the time resolution of all these different measurements (within minutes). Rather it is Fig. 2 makes the comparison of the time resolution of iSCAT measurements of CDSA. In our original manuscript we included higher time resolution measurements of platelet growth as Fig. S2. To address this comment, we now reanalyzed and moved this data to the main manuscript and updated Figs. 2 G & H.

P9.L156: “iSCAT imaging can also be used to monitor platelet growth at higher time resolution (Figs. 2G & H).....”

In addition, we have ensured that the temporal resolution of this measurement is compared to these other methods is now described in full in our introduction.

P2.L33: “Transmission electron microscopy (TEM) is arguably the most common method used to study the size evolution of individual polymer assemblies.....”

7. Page 10, First Paragraph: Author speculation for the anomalous data noted in Fig 4C is too brief. Please expand upon this supposition (evidence for 5% THF content being the point where significant solubilization occurs, or other evidence to support this claim) to better improve the real data outcomes from this study.

We have expanded on our original discussion and included previously reported results to further support our study.

P13.L207: “Fig. 4C illustrates the overall inhibitory effect of THF addition: Increasing the volume fraction of good solvent hinders the platelet growth rate, as crystallization is slowed due to enhanced polymer solubility.³⁰ ...”

P13.L213: “Similar results have been observed previously, attributing the increased final size to the solubilization of the seeds themselves at high THF content. This solubilization leads to a decrease in crystalline nuclei, thereby providing an additional source of unimers for remaining nuclei.¹⁵ ...”

8. Page 10, Second Paragraph: The observed elongation of the platelets as a function of THF concentration deserves some exploration. What is the significance of this observed fact? Why with solvent concentration? Previous observation of axial growth heterogeneity in living CDSA? All of these should be considered here even if the answer is that it’s unknown and warrants further study.

We have added this discussion in our revised manuscript.

P14.L226: “However, the mechanism leading to this asymmetry remains unknown and warrants further study. ...”

9. Page 13, Second Paragraph: The declaration of sub-millisecond temporal resolution does not seem properly supported or otherwise discussed within the main body of the work. This is particularly confusing when the stated acquisition frequency in the methods is single digit Hz. Additional context for this claim, and/or clarification of its origin should be stated/restated here

We agree, in our original manuscript we included higher time resolution measurements of platelet growth as Fig. S2. To address this comment, we now reanalyzed and moved this data to the main manuscript and updated Figs. 2 G & H. In addition, to maintain consistency, all the units for temporal resolution have been changed to milliseconds (ms).

P9.L156: “iSCAT imaging can also be used to monitor platelet growth at higher time resolution (Figs. 2G & H).....”

10. Page 5, Third Paragraph: Please clarify the use of the words “final platelet area”. Final as in the asymptote converged to in figure 2B, or an arbitrary timestep?

We agree, to avoid the confusion, we now describe it as ‘area distributions of platelets collected at the end of in situ recording’.

P8.L143: “The area distributions of platelets collected at the end of in-situ recording (Fig. S9A) were independent of the position of the platelet in the sample (Fig. S9B)....”

11. Page 8, First Paragraph: Please re-reference Gao (ref 53) when stating the assumption that parameter “B” can be held constant (as per their supplementary material).

Re-reference has been added as suggested.

P11.L184: “The area per unimer (B) is considered as a constant, linking the experimentally determined platelet area to the consumption of unimers.⁵⁶...”

12. Figure 4: A legend for each column of information would aid in re-referencing the content of the graphs much more quickly.

We agree, legends have been added to Fig. 4.

13. Figure 4: Please add language to indicate that the error bars are representing standard deviation of the area distribution in all plots (with error bars present, assuming that is true). Additionally, please state the number of platelets analyzed for each point (100) in the same statement as the standard deviation (earlier in the caption than its current form).

We have now updated our figure caption as suggested.

P12: “Figure 4: Dependence of platelet kinetics and morphology on reaction conditions..... Error bars in A, B and C represent the standard deviations of the platelet area distribution, calculated from 100 platelets for each time point...”

14. Figure 4: Either here or in the supplemental please add some explanation for how the various time points have been chosen for these plots. If done in the supplemental please reference here.

We have expended explanation in our main text as suggested.

P11.L193: “Platelets were prepared in bulk under various reaction conditions, with sample aliquots removed at predefined time points to fully capture the reaction kinetics. These samples were then characterized by iSCAT. ...”

15. Figure 5F: Please clarify the boundaries that are “less distinct, albeit still identifiable” in some form in this figure. Suggestion: a secondary series of images with rough boundaries overlaid on each image (color coded as each layer is added)

A figure with color-coded boundaries for each annulus was added into SI figure (Fig. S16A) as suggested.

P16.L261: “As shown in Fig. 5F, with no expected obvious difference in annulus thickness, the boundaries between each annulus are less distinct, albeit still identifiable (to clarify the boundaries, each annulus was color-coded as shown in Fig. S16A).....”

PS24.L897: “Figure S16: 3-annulus platelets formation studied by iSCAT...”

16. In the conclusion section, it is stated that iSCAT is limited to studies close to a surface. Although most iSCAT studies are performed in this mode for the convenience of higher mechanical stability, it is also possible to use an external reference for iSCAT. This is pointed out in Fig. 1d of Ref. 41 and used in some measurements of Ref. 43.

We have updated our manuscript to expand the discussion on the limitations of iSCAT and the potential use of an external reference to cope with this limitation with suggested references cited.

P18.L308: “Nevertheless, confocal variants of iSCAT hold the potential for 3D imaging much deeper into the sample solution.^{44,48}...”

Reviewer 1

1. Guo and co-workers provided a revised manuscript for the application of iSCAT to map 2D real time mapping of crystallization. The revised manuscript fails to take into the account and improve the 3D quantitative nature of iSCAT. Whilst I appreciate that the image quality of the revised manuscript has improved, this is a missed opportunity to address 3D quantitative refractive index (RI) measurement of iSCAT. Lateral resolutions of iSCAT is not below the diffraction limit, in fact, it is at the diffraction limit. However, iSCAT has higher degree of sensitivity to measure small optical path length difference (RI) in aqueous solutions (dielectric objects).

We fully appreciate the significance of the capability for iSCAT to return 3D information from the variation in interference between sample and reference. Similarly, we agree that information regarding 3D structure can be retrieved from this quantitative phase information and this aspect will be particularly powerful where 3D changes in particle morphology are important. We have significantly re-worked our discussion section in the manuscript to emphasize this point.

However, we must emphasize that our manuscript focuses on polymer self-assembly where 3D shape and kinetics are not important - rather we focus on the two-dimensional CDSA and growth from essentially flat platelets. There is no important information in the 3D height variation in our 2D samples. Only in experiments on multi-annulus platelets is there height variation: Here, height differences are due to the different polymer compositions, and we have chosen these compositions to generate differing 2D growth kinetics for each annulus. These height differences are due to the presence of an additional PCL₄₅-b-PDMA₃₄₈ block copolymer added as a co-mixture to growing platelets driven by CDSA. The expected height variations between these two compositions we characterize using AFM (Figs. 5D & E). The refractive indices of PCL and PDMA are also similar ($n_{\text{PCL}} = 1.476$; $n_{\text{PDMA}} = 1.430-1.456$).^{2,3} Thus, in our discussion, we attribute the observed changes in contrast in the growth of our multi-annulus 2D platelets to primarily from the designed large height variation between our two different compositions.

Given our emphasis on 2D kinetics, the small refractive index differences, and well characterized fixed heights in these samples, we have elected not to pursue 3D quantitative refractive index measurements throughout our existing experiments. However, to address directly this request by Reviewer 1 to examine 3D quantitative refractive index (RI) measurements we have conducted further experiments - where we have adapted recent defocus-based QPI methods⁴ to our multi-annulus platelet experiment. We have used these additional experiments to help us emphasize how quantitative phase imaging can enhance the difference in phase between different annuli with different compositions and/or thicknesses (Fig. S19). These experiments have helped strengthen our discussion on how 3D iSCAT might reveal useful nanoscopic information regarding the growth kinetic of more complex 3D samples.

P19.L319: "Here we have focused on 2D CDSA, but the interferometric nature of iSCAT also embeds axial information; enabling, for example, the measurement of nanoparticle axial position and size^{47,48,63-65}. As illustrated in Fig. S19, scanning along the axial direction of a two-annulus platelet reveals phase information, which can be used to infer the thickness of each annulus. However, to establish an accurate relationship between the sample thickness and the corresponding phase information, calibration is required, using samples with controlled refractive indices and varied thicknesses. We envisage iSCAT has

² Hirai et al. J. Appl. Polym. Sci. (2022) 139 42 e52925. 10.1002/app.52925;

³ Lazarova et al. J. Phys.: Conf. Ser. (2017) 794 1 012022. 10.1088/1742-6596/794/1/012022

⁴ Descloux et al. Nature Photonics (2018) 12 165. 10.1038/s41566-018-0109-4

great potential to be applied to profiling more complex CDSA processes, with three-dimensional shape at high spatio-temporal resolution⁶⁴⁻⁶⁶

P17.L258: “As iSCAT is sensitive to the change of both refractive index and path length difference between the scattered and the reference signals, variations in polymer composition jointly influence the iSCAT contrast. Based on previous studies, the refractive index difference between PCL₄₅ and PCL₄₅-b-PDMA₃₄₈ is relatively small ($n_{PCL}=1.476$; $n_{PDMA}=1.430-1.456$)^{58,59}. We interpret our contrast variation as primarily due to the increased thickness of platelet annuli formed from PCL₄₅: PCL₄₅-b-PDMA₃₄₈ as compared to pure PCL₄₅. This is supported by the AFM measurement of two-annulus platelets prepared following the same protocol (Figs. 5D & E).....”

P19.L321: “As illustrated in Fig. S19, scanning along the axial direction of a two-annulus platelet reveals phase information, which can be used to infer the thickness of each annulus. ”

PS26.L954: “Figure S19: Quantitative refractive index imaging of two-annulus platelet. ”

2. Sensitivity of iSCAT is an indication of the path length difference between the scattered light from the object and the reflected light from a reference (coverslip). Looking at Fig.5B and Fig. S16, the interference fringes (resembling Fresnel rings) are clearly visible from the iSCAT images. The authors claims that this is a multi-annulus platelet formation, but the interference fringes can also be a result of varying path-lengths (refractive index) within the platelet which is a volumetric measurement.

A strong possibility is a misunderstanding here in our description of multi-annulus platelets. This is not a 3D stack of 2D platelets, but rather a single flat platelet with distinct regions (annuli) that correspond to different compositions (PCL₄₅ vs PCL₄₅: PCL₄₅-b-PDMA₃₄₈ mixture).

We fully agree with Reviewer 1 that the interference fringes and regions of different contrast from our iSCAT images of multi-annulus platelets are a result of varying path length. This could be due to a difference in refractive index or a difference in thickness. We have deliberately selected two compositions with large differences in polymer thickness (PCL₄₅ vs. PCL₄₅: PCL₄₅-b-PDMA₃₄₈ mixture). Our AFM experiments in Figs. 5D & E confirm a significant height variation in the different annuli. Furthermore, as the refractive indices of the two polymers comprising the two different annuli regions are similar ($n_{PCL}=1.476$; $n_{PDMA}=1.430-1.456$)^{2,3} we attribute the change in contrast to be primarily a thickness effect. We have amended our description of these results to make this point clear.

P15.L249: “The growth occurs radially from the edges of the flat platelet This is supported by the AFM measurement of two-annulus platelets prepared following the same protocol, with each successive annulus extending the preceding one.....”

P17.L258: “As iSCAT is sensitive to the change of both refractive index and path length difference between the scattered and the reference signals, variations in polymer composition jointly influence the iSCAT contrast. Based on previous studies, the refractive index difference between PCL₄₅ and PCL₄₅-b-PDMA₃₄₈ is relatively small ($n_{PCL}=1.476$; $n_{PDMA}=1.430-1.456$)^{58,59}. We interpret our contrast variation as primarily due to the increased thickness of platelet annuli formed from PCL₄₅: PCL₄₅-b-PDMA₃₄₈ as compared to pure PCL₄₅. This is supported by the AFM measurement of two-annulus platelets prepared following the same protocol (Figs. 5D & E).....”

3. I would strongly urge the author to carry out additional 3D quantitative measurement that can be used to determine the refractive index changes in the platelet formation. Without additional axial measurement, it will be challenging for readers to appreciate the quantitative refractive index. In the simplest case, a background image can be recorded before the signal appears in the region of interest, e.g., before the particle or the molecule attaches to the substrate under observation. (13-16) In the cases where the object of interest remains in the field of view throughout the recording, (9, 11, 24-26) it may be possible to “extract” a background image from the recorded video by examining the data pixel by pixel. In principle, one should be able to obtain the background value of every pixel as long as the object moves far enough so that each pixel is not occupied by the object for at least one frame. In practice, it requires a sufficient number of observations of the background value of the pixel throughout the recording in order to decide which intensity value corresponds to the background. When the above condition is met, the background value can be reasonably estimated by finding the median value of the recorded intensities of individual pixels. (25) imaging of nanocrystals growth using iSCAT.

As per Q2 from this reviewer, we have conducted additional QPI measurements of multi-annulus platelets (Fig. S19) to address Reviewer 1's request for experiments that help readers to better appreciate the quantitative refractive index imaging of CDSA using iSCAT.

P19.L321: “As illustrated in Fig. S19, scanning along the axial direction of a two-annulus platelet reveals phase information, which can be used to infer the thickness of each annulus. ”

PS26.L954: “Figure S19: Phase imaging of two-annulus platelet. ”

Reviewer 2

1a&b. Fig 2C and D only included additional data (40 platelets). Whilst this increase sample size, there is lack of discussion on statistic significant. At the moment, Fig. 2C should include variance of the area over time instead of scattered plots.

In our revised Fig. 2C&D we have increased the number of platelets displayed as requested by Reviewer 2. We elected to show 40 representative platelets in Fig 2D as we think it is important to present to the reader at least some raw data for growth kinetics of 2D platelets without averaging. Adding more platelets to Fig. 2D renders the individual growth trajectories unclear.

In later figures (e.g. each subplot in Fig 4, data from ~100 platelets, we do collate data for each time point, reporting mean and s.d. for this sample). We agree that is important to describe these results in the context of their statistical significance. We now include a statistical measure of the confidence of our kinetic fits to raw data and add a discussion on statistical significance to the manuscript.

To address this request directly, we have also collated data presented in Fig 2C&D and provide plots with average values and variances (standard deviations) in Fig. S10.

P8.L143: “The individual growth trajectories of 40 representative platelets are shown in Fig. 2D, with their mean area evolution and standard deviation displayed in Fig. S10. All platelets show similar growth kinetics and as shown in Fig. S11A, at the final time point (1500 s), the platelet areas followed a Gaussian distribution with a mean of $1.12 \mu\text{m}^2$ and a standard deviation of $0.077 \mu\text{m}^2$”

P14.L207: “In addition, the coefficient of determination (R^2) reported for all concentrations indicates that the model provides a good fit.”

3. Fig. S1 attempts to discuss the background subtraction. However, the background subtraction here do not appears to be iSCAT specific (<https://pubs.acs.org/doi/full/10.1021/acsp Photonics.7b00238>). In my view, the background subtraction in iSCAT requires a more in depth estimate of background noise. This is lacking in the supplementary results that need to be done so that it is consistent with modern iSCAT systems. It is important to bear in mind that iSCAT images are interferometric images (intensity fringes) which are not standard intensities only images.

The reference suggested by Reviewer 2 (Cheng & Hsieh, ACS Photonics 2017, 4, 7, 1730–1739) describes an iterative algorithm to fit a moving PSF in the presence of a static unknown background to tackle cases where the displacement of the diffraction-limited particle is smaller than the PSF. Although clearly powerful, this method is not applicable here. Our work characterizes the 2D growth of microscopic platelets much larger than the PSF, hence we cannot simply employ Gaussian fitting of diffraction limited objects. However, this paper is also helpful in identifying the correct approach in our case:

“In the simplest case, a background image can be recorded before the signal appears in the region of interest, e.g., before the particle or the molecule attaches to the substrate under observation. (13-16) In the cases where the object of interest remains in the field of view throughout the recording, (9, 11, 24-26) it may be possible to “extract” a background image from the recorded video by examining the data pixel by pixel. In principle, one should be able to obtain the background value of every pixel as long as the object moves far enough so that each pixel is not occupied by the object for at least one frame. In practice, it requires a sufficient number of observations of the background value of the pixel throughout the recording in order to decide which intensity value corresponds to the background. When the above condition is met, the background value can be reasonably estimated by finding the median value of the recorded intensities of individual pixels. (25)”

This is precisely the case for our data, and indeed the approach we have taken.

The broader point raised here by our reviewer is highly pertinent, and we fully agree that it is important to be clear about our precise method for background subtraction and quantify this clearly. To address this, we have revised our SI description of background correction methodology to explain specific strategies for background subtraction relevant to iSCAT (ref. iv-vi in Supporting Information). We have also revised our methods description to explicitly state our image processing steps for background correction (SI methods: Data acquisition and analysis). To further quantify the background noise in our data we include directly the intermediate analysis steps used to optimize our experiment and process our data. We include a detailed assessment of noise reduction, which is now provided as Figure S2.

PS7.L896: “Background correction is a common step in iSCAT data processing used to eliminate systematic artifacts....”

PS10.L911: “To evaluate the effectiveness of our noise reduction method, we utilized the data presented in Figs. 2E and F as a demonstration.....”

4. I am able to able comment on this. Because I believe the authors have not developed more insights from iSCAT images which is axial information (thickness). As such, i think that the revised manuscript need to reflect that.

We fully appreciate the significance of the capability for iSCAT to return 3D information from the variation in interference between sample and reference. We have reflected on this and made significant changes to the discussion section of the manuscript to emphasize this point, particularly for more complex samples where the 3D structure of self-assembled particles will be important. However, we

would like to emphasize that our manuscript focuses not on polymer self-assembly where 3D shape and kinetics are important, but rather on the two-dimensional CDSA and growth of from essentially flat platelets. There is no important kinetic information in the 3D height variation in our 2D samples.

Only in our experiments on multi-annulus platelets is there height variation; here height differences are due to the different polymer compositions we have chosen to generate differing regions of 2D growth for each annulus in a flat 2D platelet.

P19.L319: “Here we have focused on 2D CDSA, but the interferometric nature of iSCAT also embeds axial information; enabling, for example, the measurement of nanoparticle axial position and size^{47,48,63-65}. As illustrated in Fig. S19, scanning along the axial direction of a two-annulus platelet reveals phase information, which can be used to infer the thickness of each annulus. However, to establish an accurate relationship between the sample thickness and the corresponding phase information, calibration is required, using samples with controlled refractive indices and varied thicknesses. We envisage iSCAT has great potential to be applied to profiling more complex CDSA processes, with three-dimensional shape at high spatio-temporal resolution⁶⁴⁻⁶⁶.....”

P17.L258: “As iSCAT is sensitive to the change of both refractive index and path length difference between the scattered and the reference signals, variations in polymer composition jointly influence the iSCAT contrast. Based on previous studies, the refractive index difference between PCL₄₅ and PCL₄₅-b-PDMA₃₄₈ is relatively small ($n_{PCL}=1.476$; $n_{PDMA}=1.430-1.456$)^{58,59}. We interpret our contrast variation as primarily due to the increased thickness of platelet annuli formed from PCL₄₅; PCL₄₅-b-PDMA₃₄₈ as compared to pure PCL₄₅. This is supported by the AFM measurement of two-annulus platelets prepared following the same protocol (Figs. 5D & E).....”

8. It is still unclear the real time aspect. What is consider real time in platform formation from seed to imaging...A imaging protocol in supplementary can help.

A detailed imaging protocol has now been added as supplementary methods to be explicit regarding the real-time nature of these measurements. In our study, "real-time" refers to the continuous measurement of the crystallization-driven self-assembly of block copolymers as it occurs. This encompasses the entire process from the moment the polymer mixture is applied to the seed-coated surface until the formation of 2D platelets is fully developed.

PS6.L872: “In this study, ‘real-time’ refers to the continuous observation of the formation process of CDSA assemblies, initiated from the block copolymer self-assemble onto the seeds on the surface and developing into 1D or 2D structures.....”

20. The authors has not provide clear discussion of Refractive index change in platelets which can affect iSCAT results.

We appreciate that any factor changing path length, be that thickness or refractive index change, will affect our iSCAT results. A detailed discussion of the impact of refractive index on iSCAT contrast has now been added to the manuscript.

We expect no variation in refractive index for single-component platelets. For multi-annulus platelets, the refractive indices of PCL and PDMA are very similar ($n_{PCL}=1.476$; $n_{PDMA}=1.430-1.456$).^{2,3} As a result, in these multi-annulus platelet we attribute differences in contrast as predominantly due to height differences from the deliberate addition of a PDMA copolymer block in one of the solutions used to grow

an annulus to growing platelets. The expected height variations between these two compositions we characterize using AFM (Figs. 5D & E).

P17.L258: “As iSCAT is sensitive to the change of both refractive index and path length difference between the scattered and the reference signals, variations in polymer composition jointly influence the iSCAT contrast. Based on previous studies, the refractive index difference between PCL₄₅ and PCL₄₅-b-PDMA₃₄₈ is relatively small ($n_{PCL}= 1.476$; $n_{PDMA}=1.430-1.456$)^{58,59}. We interpret our contrast variation as primarily due to the increased thickness of platelet annuli formed from PCL₄₅: PCL₄₅-b-PDMA₃₄₈ as compared to pure PCL₄₅. This is supported by the AFM measurement of two-annulus platelets prepared following the same protocol (Figs. 5D & E).....”

23. The authors fail to discuss about vibration isolation which is what is needed to be discussed here, particular for interferometry.

We added a specific description of how our microscope is vibrationally isolated, noting that the common-path geometry of iSCAT is much less sensitive to vibrations than conventional interferometry.

P4.L65: “Compared to conventional interferometry, which is highly sensitive to environmental factors such as vibrations and temperature changes and often requires sophisticated vibration isolation.....”

Reviewer 3

No further changes requested, supportive of manuscript in its current format.

Reviewer 4

No further changes requested, supportive of manuscript in its current format.

Reviewer 1

1. The revised manuscript clarifies that 3D information derived from iSCAT is not valuable in this study. However, the authors state, "The refractive indices of PCL and PDMA are also similar ($n_{PCL}=1.476$; $n_{PDMA}=1.430-1.456$)." This is an inaccurate statement. Interferometry, particularly with iSCAT, can detect refractive index (RI) differences as small as 0.001. Therefore, suggesting that the RI difference between PCL and PDMA is similar is an understatement. It is important to note that iSCAT measures optical path length differences because of either refractive index or thickness at the nanoscale. The real challenge lies in decoupling the effects of thickness and RI. I recommend rephrasing the statement to reflect this more accurately.

We agree that the statement "the refractive indices of PCL and PDMA are similar" was inaccurate and acknowledge that iSCAT possesses the sensitivity to detect refractive index differences, down to ~ 0.005 (though we could not find a reference supporting sensitivity as low as 0.001). We also recognize the inherent challenges in decoupling these two factors. Contrast variations between the two annuli are influenced by the combined effects of refractive index and thickness differences. Accordingly, we have revised the text in the manuscript to better reflect this complexity.

P17.L258: "This difference is likely due to a combination of both refractive index and thickness variations resulting from differences in polymer composition. As iSCAT is highly sensitive to variations in refractive index, with a detection limit as low as 0.005,^{43,58} the refractive index difference between PCL ($n=1.476$) and PDMA ($n=1.430-1.456$)^{59,60} can contribute to the observed contrast variation...."

P17.L266: "However, despite AFM providing direct evidence that thickness variations contribute significantly to the observed iSCAT contrast, it remains challenging to clearly decouple the impact of refractive index and thickness, as both factors influence the optical path length at the nanoscale."

P19.L311: "Lastly, in a multi-annulus platelet, the contrast variations between each annulus can be attributed to changes in both refractive index and thickness resulting from variations in composition. The interplay between these two factors is challenging to fully decouple without additional information."

2. Furthermore, the authors should address the practical resolution of this issue, which can be evaluated through AFM methods (as shown in their supplementary). Including this information in the main text can be valuable. will help provide a clearer explanation of the challenges involved in distinguishing between thickness and RI variations.

As suggested by Reviewer 1 the application of AFM to quantitatively evaluate the thickness of each annual has been added to the manuscript. The challenge of decoupling refractive index and thickness variations in this context has been expanded.

P17.L262: "Additionally, the increased thickness of platelet annuli formed from PCL₄₅:PCL₄₅-b-PDMA₃₄₈, compared to pure PCL₄₅, as shown in the AFM measurements of two-annulus platelets prepared following the same protocol (Figs. 5D, E & S17), can also lead to a path length difference between the scattered and reference signals. However, despite AFM providing direct evidence that thickness variations contribute significantly to the observed iSCAT contrast, it remains challenging to clearly decouple the impact of refractive index and thickness, as both factors influence the optical path length at the nanoscale."

3. The manuscript states, "This could be due to a difference in refractive index or a difference in thickness," and later clarifies, "we attribute the change in contrast to be primarily a thickness effect." However, this explanation may be causing confusion. If the effect is related to thickness, 3D measurements would indeed be necessary to resolve the variation. On the other hand, changes in refractive index could be linked to the crystallization process. To avoid this ambiguity, I urge the authors to provide a more detailed discussion of the relationship between thickness and refractive index in this context. It would be helpful to outline potential future work focused on decoupling the effects of thickness and refractive index. I also appreciate that although interferometric signals may be less quantitative, they provide valuable insight for routine nanoscale inspection.

We agree that the contrast variations are influenced by the combined effects of refractive index and thickness differences. Accordingly, we have revised the text in the manuscript to better reflect this complexity and discuss the relationship between thickness and refractive index and potential future works. Additionally, we also include a discussion that acknowledges the potential of iSCAT to provide valuable insight for routine nanoscale inspection.

P19.L311: "Lastly, in a multi-annulus platelet, the contrast variations between each annulus can be attributed to changes in both refractive index and thickness resulting from variations in composition. The interplay between these two factors is challenging to fully decouple without additional information."

P19.L313: "The interplay between these two factors is challenging to fully decouple without additional information. iSCAT's ability to provide rapid, label-free insights into structures and composition suggests significant potential for detecting relative changes in polymeric particle properties. We anticipate that further development of methods for decoupling thickness and refractive index effects will enhance the quantitative capabilities of iSCAT, leading to broader applications."

4. Figure S19 should include a colormap indicate either phase in radian or pathlength difference.

A colormap that indicates pathlength difference has been added to the revised manuscript.

PS29.L971: "Phase imaging of two-annulus platelet....."